# Learning Supervised PageRank with Gradient-Based and Gradient-Free Optimization Methods

**Lev Bogolubsky**[1,2]**, Gleb Gusev**[1,5]**, Andrei Raigorodskii**[5,2,1,8]**, Aleksey Tikhonov**[1]**, Maksim Zhukovskii**[1,5]
Yandex[1], Moscow State University[2], Buryat State University[8]
`{bogolubsky, gleb57, raigorodsky, altsoph, zhukmax}@yandex-team.ru`

**Pavel Dvurechensky**[3,4]**, Alexander Gasnikov**[4,5]
Weierstrass Institute[3], Institute for Information Transmission Problems RAS[4],
Moscow Institute of Physics and Technology[5]
`pavel.dvurechensky@wias-berlin.de, gasnikov@yandex.ru`

**Yurii Nesterov**[6,7]
Center for Operations Research and Econometrics[6],
Higher School of Economics[7]
`yurii.nesterov@uclouvain.be`

## Abstract

In this paper, we consider a non-convex loss-minimization problem of learning Supervised PageRank models, which can account for features of nodes and edges. We propose gradient-based and random gradient-free methods to solve this problem. Our algorithms are based on the concept of an inexact oracle and unlike the state-of-the-art gradient-based method we manage to provide theoretically the convergence rate guarantees for both of them. Finally, we compare the performance of the proposed optimization methods with the state of the art applied to a ranking task.

## 1 INTRODUCTION

The most acknowledged methods of measuring importance of nodes in graphs are based on random walk models. Particularly, PageRank [18], HITS [11], and their variants [8, 9, 19] are originally based on a discrete-time Markov random walk on a link graph. Despite undeniable advantages of PageRank and its mentioned modifications, these algorithms miss important aspects of the graph that are not described by its structure. In contrast, a number of approaches allows to account for different properties of nodes and edges between them by encoding them in restart and transition probabilities (see [3, 4, 6, 10, 12, 20, 21]). These properties may include, e.g., the statistics about users' interactions with the nodes (in web graphs [12] or graphs of social networks [2]), types of edges (such as URL redirecting in web graphs [20]) or histories of nodes' and edges' changes [22].

In the general ranking framework called Supervised PageRank [21], weights of nodes and edges in a graph are linear combinations of their features with coefficients as the model parameters. The existing optimization method [21] of learning these parameters and the optimizations methods proposed in the presented paper have two levels. On the lower level, the following problem is solved: to estimate the value of the loss function (in the case of zero-order oracle) and its derivatives (in the case of first-order oracle) for a given parameter vector. On the upper level, the estimations obtained on the lower level of the optimization methods (which we also call inexact oracle information) are used for tuning the parameters by an iterative algorithm. Following [6], the authors of Supervised PageRank consider a non-convex loss-minimization problem for learning the parameters and solve

it by a two-level gradient-based method. On the lower level of this algorithm, an estimation of the stationary distribution of the considered Markov random walk is obtained by classical power method and estimations of derivatives w.r.t. the parameters of the random walk are obtained by power method introduced in [23, 24]. On the upper level, the obtained gradient of the stationary distribution is exploited by the gradient descent algorithm. As both power methods give imprecise values of the stationary distribution and its derivatives, there was no proof of the convergence of the state-of-the-art gradient-based method to a stationary point.

The considered non-convex loss-minimization problem [21] can not be solved by existing optimization methods such as [16] and [7] due to presence of constraints for parameter vector and the impossibility to calculate the exact value of the loss function. Moreover, standard global optimization methods can not be applied, because they require unbiased estimations of the loss function.

In our paper, we propose two two-level methods to solve the problem [21]. On the lower level of these methods, we use the linearly convergent method [17] to calculate an approximation to the stationary distribution of Markov random walk. We show that this method allows to approximate the value of the loss function at any given accuracy and has the lowest proved complexity bound among methods proposed in [5]. We develop a gradient method for general constrained non-convex optimization problems with inexact oracle, estimate its convergence rate to the stationary point of the problem. We exploit this gradient method on the upper level of the two-level algorithm for learning Supervised PageRank. Our contribution to the gradient-free methods framework consists in adapting the approach of [16] to the case of constrained optimization problems when the value of the function is calculated with some known accuracy. We prove a convergence theorem for this method and exploit it on the upper level of the second two-level algorithm.

Another contribution consists in investigating both for the gradient and gradient-free methods the trade-off between the accuracy of the lower-level algorithm, which is controlled by the number of iterations of method in [17] and its generalization (for derivatives estimation), and the computational complexity of the two-level algorithm as a whole. Finally, we estimate the complexity of the whole two-level algorithms for solving the loss-minimization problem with a given accuracy.

In the experiments, we apply our algorithms to learning Supervised PageRank on a real ranking task. Summing up, both two-level methods, unlike the state-of-the-art [21], have theoretical guarantees on convergence rate, and outperform it in the ranking quality in experiments. The main advantages of the first gradient-based algorithm: the guarantees of a convergence do not require the convexity, this algorithm has less input parameters than gradient-free one. The main advantage of the second gradient-free algorithm is that it avoids calculating the derivative for each element of a large matrix.

## 2  MODEL DESCRIPTION

We concider the following random walk on a directed graph $\Gamma = (V, E)$ introduced in [21]. Assume that each node $i \in V$ and each edge $i \to j \in E$ is represented by a vector of features $\mathbf{V}_i \in \mathbb{R}_+^{m_1}$ and a vector of features $\mathbf{E}_{ij} \in \mathbb{R}_+^{m_2}$ respectively. A surfer starts from a random page $v_0$ of a *seed set* $U \subset V$. The *restart probability* that $v_0 = i$ equals

$$[\pi^0]_i = \frac{\langle \varphi_1, \mathbf{V}_i \rangle}{\sum_{l \in U} \langle \varphi_1, \mathbf{V}_l \rangle}, \quad i \in U \tag{2.1}$$

and $[\pi^0]_i = 0$ for $i \in V \setminus U$, where $\varphi_1 \in \mathbb{R}^{m_1}$ is a parameter, which conducts the random walk. We assume that $\sum_{l \in U} \langle \varphi_1, \mathbf{V}_l \rangle$ should be non-zero.

At each step, the surfer makes a *restart* with probability $\alpha \in (0, 1)$ (originally [18], $\alpha = 0.15$) or traverses an outgoing edge (makes a *transition*) with probability $1 - \alpha$. In the former case, the surfer chooses a vertex according to the distribution $\pi^0$. In the latter one, the *transition probability* of traversing an edge $i \to j \in E$ is

$$[P]_{i,j} = \frac{\langle \varphi_2, \mathbf{E}_{ij} \rangle}{\sum_{l:i \to l} \langle \varphi_2, \mathbf{E}_{il} \rangle}, \tag{2.2}$$

where $\varphi_2 \in \mathbb{R}^{m_2}$ is a parameter and the current position $i$ has non-zero outdegree, and $[P(\varphi)]_{i,j} = [\pi^0(\varphi)]_j$ for all $j \in V$ if the outdegree of $i$ is zero (thus the surfer always makes a restart in this case). We assume that $\sum_{l:i \to l} \langle \varphi_2, \mathbf{E}_{il} \rangle$ is non-zero for all $i$ with non-zero outdegree.

By Equations 2.1 and 2.2, the total probability of choosing vertex $j \in V$ conditioned by the surfer being at vertex $i$ equals $\alpha[\pi^0(\varphi)]_j + (1 - \alpha)[P(\varphi)]_{i,j}$, where $\varphi = (\varphi_1, \varphi_2)^T$ and we use $\pi^0(\varphi)$ and $P(\varphi)$ to express the dependence of $\pi^0, P$ on the parameters.

The stationary distribution $\pi(\varphi) \in \mathbb{R}^p$ of the described Markov process is a solution of the system

$$\pi = \alpha\pi^0(\varphi) + (1 - \alpha)P^T(\varphi)\pi. \tag{2.3}$$

In this paper, we learn an algorithm, which ranks nodes $i$ according to scores $[\pi(\varphi)]_i$.

Let $Q$ be a set of queries and a set of nodes $V_q \subset V$ is associated to each query $q$. For example, vertices in $V_q$ may represent web pages visited by users after submitting query $q$. For each $q \in Q$, some nodes of $V_q$ are manually judged by relevance labels $1, \ldots, \ell$. Our goal is to learn the parameter vector $\varphi$ of a ranking algorithm $\pi_q = \pi_q(\varphi)$ which minimizes the discrepancy of its ranking scores $[\pi_q]_i$, $i \in V_q$, from the the assigned labels. We consider the square loss function [12, 21, 22]

$$f(\varphi) = \frac{1}{|Q|}\sum_{q=1}^{|Q|}\|(A_q\pi_q(\varphi))_+\|_2^2. \tag{2.4}$$

Each row of matrix $A_q \in \mathbb{R}^{r_q \times p_q}$ corresponds to some pair of pages $i_1, i_2 \in V_q$ such that the label of $i_1$ is strictly greater than the label of $i_2$ (we denote by $r_q$ the number of all such pairs from $V_q$ and $p_q := |V_q|$). The $i_1$-th element of this row is equal to $-1$, $i_2$-th element is equal to $1$, and all other elements are equal to 0. Vector $x_+$ has components $[x_+]_i = \max\{x_i, 0\}$.

To make ranking scores (2.3) query–dependent, we assume that $\pi$ is defined on a query–dependent graph $\Gamma_q = (V_q, E_q)$ with query-dependent feature vectors $\mathbf{V}_i^q$, $i \in V_q$, $\mathbf{E}_{ij}^q$, $i \to j \in E_q$. For example, these features may reflect different aspects of query–page relevance. For a given $q \in Q$, we consider all the objects related to the graph $\Gamma_q$ introduced above: $U_q := U$, $\pi_q^0 := \pi^0$, $P_q := P$, $\pi_q := \pi$. In this way, the ranking scores $\pi_q$ depend on query via the query–dependent features, but the parameters of the model $\alpha$ and $\varphi$ are not query–dependent. In what follows, we use the following notations throughout the paper: $n_q := |U_q|$, $m = m_1 + m_2$, $r = \max_{q \in Q} r_q$, $p = \max_{q \in Q} p_q$, $n = \max_{q \in Q} n_q$, $s = \max_{q \in Q} s_q$, where $s_q = \max_{i \in V_q}|\{j : i \to j \in E_q\}|$. In order to guarantee that the probabilities in (2.1) and (2.2) are correctly defined, we need to appropriately choose a set $\Phi$ of possible values of parameters $\varphi$. We choose some $\hat{\varphi}$ and $R > 0$ such that $\Phi = \{\varphi \in \mathbb{R}^m : \|\varphi - \hat{\varphi}\|_2 \leq R\}$ lies in the set of vectors with positive components $\mathbb{R}_{++}^m$ [1]. In this paper, we solve the following loss-minimization problem:

$$\min_{\varphi \in \Phi} f(\varphi), \Phi = \{\varphi \in \mathbb{R}^m : \|\varphi - \hat{\varphi}\|_2 \leq R\}. \tag{2.5}$$

# 3  NUMERICAL CALCULATION OF $f(\varphi)$ AND $\nabla f(\varphi)$

Our goal is to provide methods for solving Problem 2.5 with guarantees on rate of convergence and complexity bounds. The calculation of the values of $f(\varphi)$ and its gradient $\nabla f(\varphi)$ is problematic, since it requires to calculate those for $|Q|$ vectors $\pi_q(\varphi)$ defined by Equation 2.3. While the exact values are impossible to derive in general, existing methods provide estimations of $\pi_q(\varphi)$ and its derivatives $\frac{d\pi_q(\varphi)}{d\varphi^T}$ in an iterative way with a trade-off between time and accuracy. To be able to guarantee convergence of our optimization algorithm in this inexact oracle setting, we consider numerical methods that calculate approximation for $\pi_q(\varphi)$ and its derivatives with any required accuracy. We have analysed state-of-the-art methods summarized in the review [5] and power method used in [18, 2, 21] and have found that the method [17] is the most suitable.

It constructs a sequence $\pi_k$ and outputs $\tilde{\pi}_q(\varphi, N)$ by the following rule (integer $N > 0$ is a parameter):

$$\pi_0 = \pi_q^0(\varphi), \quad \pi_{k+1} = P_q^T(\varphi)\pi_k, \quad \tilde{\pi}_q(\varphi, N) = \frac{\alpha}{1 - (1 - \alpha)^{N+1}}\sum_{k=0}^{N}(1 - \alpha)^k\pi_k. \tag{3.1}$$

**Lemma 1.** *Assume that, for some $\delta_1 > 0$, Method 3.1 with $N = \left\lceil \frac{1}{\alpha} \ln \frac{8r}{\delta_1} \right\rceil - 1$ is used to calculate the vector $\tilde{\pi}_q(\varphi, N)$ for every $q \in Q$. Then $\tilde{f}(\varphi, \delta_1) = \frac{1}{|Q|} \sum_{q=1}^{|Q|} \|(A_q \tilde{\pi}_q(\varphi, N))_+\|_2^2$ satisfies $|\tilde{f}(\varphi, \delta_1) - f(\varphi)| \leq \delta_1$. Moreover, the calculation of $\tilde{f}(\varphi, \delta_1)$ requires not more than $|Q|(3mps + 3psN + 6r)$ a.o.*

The proof of Lemma 1 is in Supplementary Materials.

Let $p_i(\varphi)$ be the $i$-th column of the matrix $P_q^T(\varphi)$. Our generalization of the method [17] for calculation of $\frac{d\pi_q(\varphi)}{d\varphi^T}$ for any $q \in Q$ is the following. Choose some non-negative integer $N_1$ and calculate $\tilde{\pi}_q(\varphi, N_1)$ using (3.1). Choose some $N_2 \geq 0$, calculate $\Pi_k$, $k = 0, ..., N_2$ and $\tilde{\Pi}_q(\varphi, N_2)$

$$\Pi_0 = \alpha \frac{d\pi_q^0(\varphi)}{d\varphi^T} + (1-\alpha) \sum_{i=1}^{p_q} \frac{dp_i(\varphi)}{d\varphi^T} [\tilde{\pi}_q(\varphi, N_1)]_i, \quad \Pi_{k+1} = P_q^T(\varphi)\Pi_k, \tag{3.2}$$

$$\tilde{\Pi}_q(\varphi, N_2) = \frac{1}{1 - (1-\alpha)^{N_2+1}} \sum_{k=0}^{N_2} (1-\alpha)^k \Pi_k. \tag{3.3}$$

In what follows, we use the following norm on the space of matrices $A \in \mathbb{R}^{n_1 \times n_2}$: $\|A\|_1 = \max_{j=1,...,n_2} \sum_{i=1}^{n_1} |a_{ij}|$.

**Lemma 2.** *Let $\beta_1$ be some explicitly computable constant (see Supplementary Materials). Assume that Method 3.1 with $N_1 = \left\lceil \frac{1}{\alpha} \ln \frac{24\beta_1 r}{\alpha\delta_2} \right\rceil - 1$ is used for every $q \in Q$ to calculate the vector $\tilde{\pi}_q(\varphi, N_1)$ and Method 3.2, 3.3 with $N_2 = \left\lceil \frac{1}{\alpha} \ln \frac{8\beta_1 r}{\alpha\delta_2} \right\rceil - 1$ is used for every $q \in Q$ to calculate the matrix $\tilde{\Pi}_q(\varphi, N_2)$ (3.3). Then the vector $\tilde{g}(\varphi, \delta_2) = \frac{2}{|Q|} \sum_{q=1}^{|Q|} \left(\tilde{\Pi}_q(\varphi, N_2)\right)^T A_q^T (A_q \tilde{\pi}_q(\varphi, N_1))_+$ satisfies $\|\tilde{g}(\varphi, \delta_2) - \nabla f(\varphi)\|_\infty \leq \delta_2$. Moreover, the calculation of $\tilde{g}(\varphi, \delta_2)$ requires not more than $|Q|(10mps + 3psN_1 + 3mpsN_2 + 7r)$ a.o.*

The proof of Lemma 2 can be found in Supplementary Materials.

## 4 RANDOM GRADIENT-FREE OPTIMIZATION METHODS

In this section, we first describe general framework of random gradient-free methods with inexact oracle and then apply it for Problem 2.5. Lemma 1 allows to control the accuracy of the inexact zero-order oracle and hence apply random gradient-free methods with inexact oracle.

### 4.1 GENERAL FRAMEWORK

Below we extend the framework of random gradient-free methods [1, 16, 7] for the situation of presence of uniformly bounded error of unknown nature in the value of an objective function in general optimization problem. Unlike [16], we consider a constrained optimization problem and a randomization on a Euclidean sphere which seems to give better large deviations bounds and doesn't need the assumption that the objective function can be calculated at any point of $\mathbb{R}^m$.

Let $\mathcal{E}$ be a $m$-dimensional vector space and $\mathcal{E}^*$ be its dual. In this subsection, we consider a general function $f(\cdot) : \mathcal{E} \to \mathbb{R}$ and denote its argument by $x$ or $y$ to avoid confusion with other sections. We denote the value of linear function $g \in \mathcal{E}^*$ at $x \in \mathcal{E}$ by $\langle g, x \rangle$. We choose some norm $\|\cdot\|$ in $\mathcal{E}$ and say that $f \in C_L^{1,1}(\|\cdot\|)$ iff $|f(x) - f(y) - \langle \nabla f(y), x-y \rangle| \leq \frac{L}{2}\|x-y\|^2, \quad \forall x, y \in \mathcal{E}$. The problem of our interest is to find $\min_{x \in X} f(x)$, where $f \in C_L^{1,1}(\|\cdot\|)$, $X$ is a closed convex set and there exists a number $D \in (0, +\infty)$ such that $\mathrm{diam}X := \max_{x,y \in X} \|x-y\| \leq D$. Also we assume that the inexact zero-order oracle for $f(x)$ returns a value $\tilde{f}(x, \delta) = f(x) + \tilde{\delta}(x)$, where $\tilde{\delta}(x)$ is the error satisfying for some $\delta > 0$ (which is known) $|\tilde{\delta}(x)| \leq \delta$ for all $x \in X$. Let $x^* \in \arg\min_{x \in X} f(x)$. Denote $f^* = \min_{x \in X} f(x)$.

Unlike [16], we define the biased gradient-free oracle $g_\tau(x, \delta) = \frac{m}{\tau}(\tilde{f}(x + \tau\xi, \delta) - \tilde{f}(x, \delta))\xi$, where $\xi$ is a random vector uniformly distributed over the unit sphere $\mathcal{S} = \{t \in \mathbb{R}^m : \|t\|_2 = 1\}$, $\tau$ is a smoothing parameter.

---

**Algorithm 1** Gradient-type method

---

**Input:** Point $x_0 \in X$, stepsize $h > 0$, number of steps $M$.
Set $k = 0$.
**repeat**
 Generate $\xi_k$ and calculate corresponding $g_\tau(x_k, \delta)$.
 Calculate $x_{k+1} = \Pi_X(x_k - hg_\tau(x_k, \delta))$ ($\Pi_X(\cdot)$ – Euclidean projection onto the set $X$).
 Set $k = k + 1$.
**until** $k > M$
**Output:** The point $y_M = \arg\min_x\{f(x) : x \in \{x_0, \ldots, x_M\}\}$.

---

**Theorem 1.** *Let $f \in C_L^{1,1}(\|\cdot\|_2)$ and convex. Assume that $x^* \in \text{int} X$, and the sequence $x_k$ is generated by Algorithm 1 with $h = \frac{1}{8mL}$. Then for any $M \geq 0$, we have $\mathbb{E}_{\Xi_{M-1}} f(y_M) - f^* \leq \frac{8mLD^2}{M+1} + \frac{\tau^2 L(m+8)}{8} + \frac{\delta m D}{4\tau} + \frac{\delta^2 m}{L\tau^2}$. Here $\Xi_k = (\xi_0, \ldots, \xi_k)$ is the history of realizations of the vector $\xi$.*

The full proof of the theorem is in Supplementary Materials.

## 4.2 SOLVING THE LEARNING PROBLEM

Now, we apply the results of Subsection 4.1 to solve Problem 2.5. Note that presence of constraints and oracle inexactness do not allow to directly apply the results of [16]. We assume that there is a local minimum $\varphi^*$, and $\Phi$ is a small vicinity of $\varphi^*$, in which $f(\varphi)$ (2.4) is convex (generally speaking, it is nonconvex). We choose the desired accuracy $\varepsilon$ for $f^*$ (the optimal value) approximation in the sense that $\mathbb{E}_{\Xi_{M-1}} f(y_M) - f^* \leq \varepsilon$. In accordance with Theorem 1, $\varepsilon$ gives the number of steps $M$ of Algorithm 1, the value of $\tau$, the value of the required accuracy $\delta$ of the inexact zero-order oracle. The value $\delta$, by Lemma 1, gives the number of steps $N$ of Method 3.1 required to calculate a $\delta$-approximation $\tilde{f}(\varphi, \delta)$ for $f(\varphi)$. Then the inexact zero-order oracle $\tilde{f}(\varphi, \delta)$ is used to make Algorithm 1 step. Theorem 1 and the choice of the feasible set $\Phi$ to be a Euclidean ball make it natural to choose $\|\cdot\|_2$-norm in the space $\mathbb{R}^m$ of parameter $\varphi$. It is easy to see that in this norm $\text{diam}\Phi \leq 2R$. Algorithm 2 in Supplementary Materials is a formal record of these ideas.

The most computationally hard on each iteration of the main cycle of this method are calculations of $\tilde{f}(\varphi_k + \tau\xi_k, \delta)$, $\tilde{f}(\varphi_k, \delta)$. Using Lemma 1, we obtain the complexity of each iteration and the following result, which gives the complexity of Algorithm 2.

**Theorem 2.** *Assume that the set $\Phi$ in (2.5) is chosen in a way such that $f(\varphi)$ is convex on $\Phi$ and some $\varphi^* \in \arg\min_{\varphi \in \Phi} f(\varphi)$ belongs also to $\text{int}\Phi$. Then the mean total number of arithmetic operations of the Algorithm 2 for the accuracy $\varepsilon$ (i.e. for the inequality $\mathbb{E}_{\Xi_{M-1}} f(\hat{\varphi}_M) - f(\varphi^*) \leq \varepsilon$ to hold) is not more than*

$$768mps|Q|\frac{LR^2}{\varepsilon}\left(m + \frac{1}{\alpha}\ln\frac{128mrR\sqrt{L(m+8)}}{\varepsilon^{3/2}\sqrt{2}} + 6r\right).$$

## 5 GRADIENT-BASED OPTIMIZATION METHODS

In this section, we first develop a general framework of gradient methods with inexact oracle for non-convex problems from rather general class and then apply it for the particular Problem 2.5. Lemma 1 and Lemma 2 allow to control the accuracy of the inexact first-order oracle and hence apply proposed framework.

## 5.1 GENERAL FRAMEWORK

In this subsection, we generalize the approach in [7] for constrained non-convex optimization problems. Our main contribution consists in developing this framework for an inexact first-order oracle and unknown "Lipschitz constant" of this oracle.

We consider a *composite optimization* problem of the form $\min_{x \in X}\{\psi(x) := f(x) + h(x)\}$, where $X \subset \mathcal{E}$ is a closed convex set, $h(x)$ is a simple convex function, e.g. $\|x\|_1$. We assume that $f(x)$ is

a general function endowed with an inexact first-order oracle in the following sense. There exists a number $L \in (0, +\infty)$ such that for any $\delta \geq 0$ and any $x \in X$ one can calculate $\tilde{f}(x, \delta) \in \mathbb{R}$ and $\tilde{g}(x, \delta) \in \mathcal{E}^*$ satisfying

$$|f(y) - (\tilde{f}(x, \delta) - \langle \tilde{g}(x, \delta), y - x \rangle)| \leq \frac{L}{2} \|x - y\|^2 + \delta. \tag{5.1}$$

for all $y \in X$. The constant $L$ can be considered as "Lipschitz constant" because for the exact first-order oracle for a function $f \in C_L^{1,1}(\| \cdot \|)$ Inequality 5.1 holds with $\delta = 0$. This is a generalization of the concept of $(\delta, L)$-oracle considered in [25] for convex problems.

We choose a *prox-function* $d(x)$ which is continuously differentiable and 1-strongly convex on $X$ with respect to $\| \cdot \|$. This means that for any $x, y \in X$ $d(y) - d(x) - \langle \nabla d(x), y - x \rangle \geq \frac{1}{2} \|y - x\|^2$. We define also the corresponding *Bregman distance* $V(x, z) = d(x) - d(z) - \langle \nabla d(z), x - z \rangle$.

---

**Algorithm 2** Adaptive projected gradient algorithm

---

**Input:** Point $x_0 \in X$, number $L_0 > 0$.
Set $k = 0$, $z = +\infty$.
**repeat**
    Set $M_k = L_k$, flag $= 0$.
    **repeat**
        Set $\delta = \frac{\varepsilon}{16 M_k}$. Calculate $\tilde{f}(x_k, \delta)$ and $\tilde{g}(x_k, \delta)$.
        Find $w_k = \arg\min_{x \in Q} \{\langle \tilde{g}(x_k, \delta), x \rangle + M_k V(x, x_k) + h(x)\}$ and calculate $\tilde{f}(w_k, \delta)$.
        If the inequality $\tilde{f}(w_k, \delta) \leq \tilde{f}(x_k, \delta) + \langle \tilde{g}(x_k, \delta), w_k - x_k \rangle + \frac{M_k}{2} \|w_k - x_k\|^2 + \frac{\varepsilon}{8 M_k}$ holds,
        set flag $= 1$. Otherwise set $M_k = 2 M_k$.
    **until** flag $= 1$
    Set $x_{k+1} = w_k$, $L_{k+1} = \frac{M_k}{2}$.
    If $\|M_k(x_k - x_{k+1})\| < z$, set $z = \|M_k(x_k - x_{k+1})\|$, $K = k$.
    Set $k = k + 1$.
**until** $z \leq \varepsilon$
**Output:** The point $x_{K+1}$.

---

**Theorem 3.** *Assume that $f(x)$ is endowed with the inexact first-order oracle in a sense* (5.1) *and that there exists a number $\psi^* > -\infty$ such that $\psi(x) \geq \psi^*$ for all $x \in X$. Then after $M$ iterations of Algorithm 2 it holds that $\|M_K(x_K - x_{K+1})\|^2 \leq \frac{4L(\psi(x_0) - \psi^*)}{M+1} + \frac{\varepsilon}{2}$. Moreover, the total number of inexact oracle calls is not more than $2M + 2\log_2 \frac{2L}{L_0}$.*

The full proof of the theorem is in Supplementary Materials.

## 5.2 SOLVING THE LEARNING PROBLEM

In this subsection, we return to Problem 2.5 and apply the results of the previous subsection. Note that we can not directly apply the results of [7] due to the inexactness of the oracle. For this problem, $h(\cdot) \equiv 0$. It is easy to show that in 1-norm $\text{diam}\Phi \leq 2R\sqrt{m}$. For any $\delta > 0$, Lemma 1 with $\delta_1 = \frac{\delta}{2}$ allows us to obtain $\tilde{f}(\varphi, \delta_1)$ such that inequality $|\tilde{f}(\varphi, \delta_1) - f(\varphi)| \leq \delta_1$ holds and Lemma 2 with $\delta_2 = \frac{\delta}{4R\sqrt{m}}$ allows us to obtain $\tilde{g}(\varphi, \delta_2)$ such that inequality $\|\tilde{g}(\varphi, \delta_2) - \nabla f(\varphi)\|_\infty \leq \delta_2$ holds. Similar to [25], since $f \in C_L^{1,1}(\| \cdot \|_2)$, these two inequalities lead to Inequality 5.1 for $\tilde{f}(\varphi, \delta_1)$ in the role of $\tilde{f}(x, \delta)$, $\tilde{g}(\varphi, \delta_2)$ in the role of $\tilde{g}(x, \delta)$ and $\| \cdot \|_2$ in the role of $\| \cdot \|$.

We choose the desired accuracy $\varepsilon$ for approximating the stationary point of Problem 2.5. This accuracy gives the required accuracy $\delta$ of the inexact first-order oracle for $f(\varphi)$ on each step of the inner cycle of the Algorithm 2. Knowing the value $\delta_1 = \frac{\delta}{2}$ and using Lemma 1, we choose the number of steps $N$ of Method 3.1 and thus approximate $f(\varphi)$ with the required accuracy $\delta_1$ by $\tilde{f}(\varphi, \delta_1)$. Knowing the value $\delta_2 = \frac{\delta}{4R\sqrt{m}}$ and using Lemma 2, we choose the number of steps $N_1$ of Method 3.1 and the number of steps $N_2$ of Method 3.2, 3.3 and obtain the approximation $\tilde{g}(\varphi, \delta_2)$ of $\nabla f(\varphi)$ with the required accuracy $\delta_2$. Then we use the inexact first-order oracle $(\tilde{f}(\varphi, \delta_1), \tilde{g}(\varphi, \delta_2))$ to perform a step of Algorithm 2. Since $\Phi$ is the Euclidean ball, it is natural to set $\mathcal{E} = R^m$ and $\| \cdot \| = \| \cdot \|_2$,

choose the prox-function $d(\varphi) = \frac{1}{2}\|\varphi\|_2^2$. Then the Bregman distance is $V(\varphi, \omega) = \frac{1}{2}\|\varphi - \omega\|_2^2$. Algorithm 4 in Supplementary Materials is a formal record of the above ideas.

The most computationally consuming operations of the inner cycle of Algorithm 4 are calculations of $\tilde{f}(\varphi_k, \delta_1)$, $\tilde{f}(\omega_k, \delta_1)$ and $\tilde{g}(\varphi_k, \delta_2)$. Using Lemma 1 and Lemma 2, we obtain the complexity of each iteration. From Theorem 3 we obtain the following result, which gives the complexity of Algorithm 4.

**Theorem 4.** *The total number of arithmetic operations in Algorithm 4 for the accuracy $\varepsilon$ (i.e. for the inequality $\|M_K(\varphi_K - \varphi_{K+1})\|_2^2 \leq \varepsilon$ to hold) is not more than*

$$\left( \frac{8L(f(\varphi_0) - f^*)}{\varepsilon} + \log_2 \frac{2L}{L_0} \right) \cdot \left( 7r|Q| + \frac{6mps|Q|}{\alpha} \ln \frac{1024\beta_1 r R L\sqrt{m}}{\alpha\varepsilon} \right).$$

# 6 EXPERIMENTAL RESULTS

In this section, we compare our gradient-free and gradient-based methods with the state-of-the-art gradient-based method [21] on the web page ranking problem. In the next section, we describe the dataset. In Section 6.2, we report the results of the experiments.

## 6.1 DATA

We consider the user web browsing graph $\Gamma_q = (V_q, E_q)$, $q \in Q$, introduced in [12]. Unlike a link graph, a user browsing graph is query–dependent. The set of vertices $V_q$ consists of all different pages visited by users during their sessions started from $q$. The set of directed edges $E_q$ represents all the ordered pairs of neighboring elements $(\tilde{i}, i)$ from such sessions. We add a page $i$ in the seed set $U_q$ if and only if there is a session where $i$ is the first page visited after submitting query $q$.

All experiments are performed with data of a popular commercial search engine Yandex[2]. We chose a random set of 600 queries $Q$ and collected user sessions started with them. There are $\approx 11.7$K vertices and $\approx 7.5$K edges in graphs $\Gamma_q$, $q \in Q$, in total. For each query, a set of pages was labelled by professional assessors with standard 5 relevance grades ($\approx 1.7$K labeled query–document pairs in total). We divide our data into two parts. On the first part $Q_1$ (50% of the set of queries $Q$) we train the parameters and on the second part $Q_2$ we test the algorithms. For each $q \in Q$ and $i \in V_q$, vector $\mathbf{V}_i^q$ of size $m_1 = 26$ encodes features for query–document pair $(q, i)$. Vector $\mathbf{E}_{ii}^q$ of $m_2 = 52$ features for an edge $\tilde{i} \to i \in E_q$ is obtained as the concatenation of $\mathbf{V}_{\tilde{i}}^q$ and $\mathbf{V}_i^q$.

To study a dependency between the efficiency of the algorithms and the sizes of the graphs, we sort the sets $Q_1, Q_2$ in ascending order of sizes of the respective graphs. Sets $Q_j^1, Q_j^2, Q_j^3$ contain first (in terms of these order) $100, 200, 300$ elements respectively for $j \in \{1, 2\}$.

## 6.2 PERFORMANCES OF THE OPTIMIZATION ALGORITHMS

We optimized the parameters $\varphi$ by three methods: our gradient-free method GFN (Algorithm 2), the gradient-based method GBN (Algorithm 4), and the state-of-the-art gradient-method GBP. The values of hyperparameters are the following: the Lipschitz constant $L = 10^{-4}$ in GFN (and $L_0 = 10^{-4}$ in GBN), the accuracy $\varepsilon = 10^{-6}$ (in both GBN and GFN), the radius $R = 0.99$ (in both GBN and GFN). On all sets of queries, we compare final values of the loss function for GBN when $L_0 \in \{10^{-4}, 10^{-3}, 10^{-2}, 10^{-1}, 1\}$. The differences are less than $10^{-7}$. We choose $L$ in GFN to be equal to $L_0$ (we show how the choice of $L$ influences the output of the gradient-free algorithm, see supplementary materials, Figure 2). Moreover, we evaluate both our gradient-based and gradient-free algorithms for different values of the accuracies. The outputs of the algorithms differ insufficiently on all test sets $Q_2^i$, $i \in \{1, 2, 3\}$, when $\varepsilon \leq 10^{-6}$. On the lower level of the state-of-the-art gradient-based algorithm, the stochastic matrix and its derivative are raised to the power 100. We evaluate GBP for different values of the step size (50, 100, 200, 500). We stop the GBP algorithms when the differences between the values of the loss function on the next step and the current step are less than $-10^{-5}$ on the test sets.

In Table 1, we present the performances of the optimization algorithms in terms of the loss function $f$ (2.4). We also compare the algorithms with the untuned Supervised PageRank ($\varphi = \varphi_0 = e_m$). On Figure 1, we give the outputs of the optimization algorithms on each iteration of the upper levels of the learning processes on the test set $Q_2^3$, similar results were obtained for the sets $Q_2^1, Q_2^2$.

| Meth. | $Q_2^1$ | | $Q_2^2$ | | $Q_2^3$ | |
|---|---|---|---|---|---|---|
| | loss | steps | loss | steps | loss | steps |
| PR | .00357 | 0 | .00354 | 0 | .0033 | 0 |
| GBN | .00279 | 12 | .00305 | 12 | .00295 | 12 |
| GFN | .00274 | $10^6$ | .00297 | $10^6$ | .00292 | $10^6$ |
| GBP 50s. | .00282 | 16 | .00307 | 31 | .00295 | 40 |
| GBP 100s. | .00282 | 8 | .00307 | 16 | .00295 | 20 |
| GBP 200s. | .00283 | 4 | .00308 | 7 | .00295 | 9 |
| GBP 500s. | .00283 | 2 | .00308 | 2 | .00295 | 3 |

Table 1: Comparison of the algorithms on the test sets.

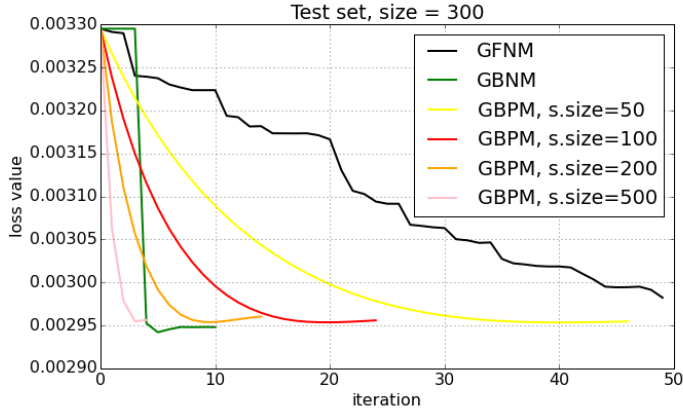

Figure 1: Values of the loss function on each iteration of the optimization algorithms on the test set $Q_2^3$.

GFN significantly outperforms the state-of-the-art algorithms on all test sets. GBN significantly outperforms the state-of-the-art algorithm on $Q_2^1$ (we obtain the $p$-values of the paired $t$-tests for all the above differences on the test sets of queries, all these values are less than 0.005). However, GBN requires less iterations of the upper level (until it stops) than GBP for step sizes 50 and 100 on $Q_2^2, Q_2^3$. Finally, we show that Nesterov–Nemirovski method converges to the stationary distribution faster than the power method (in supplementary materials, on Figure 2, we demonstrate the dependencies of the value of the loss function on $Q_1^1$ for both methods of computing the untuned Supervised PageRank $\varphi = \varphi_0 = e_m$).

## 7 CONCLUSION

We propose a gradient-free optimization method for general convex problems with inexact zero-order oracle and an adaptive gradient method for possibly nonconvex general composite optimization problems with inexact first-order oracle. For both methods, we provide convergence rate analysis. We also apply our new methods for known problem of learning a web-page ranking algorithm. Our new algorithms not only outperform existing algorithms, but also are guaranteed to solve this learning problem. In practice, this means that these algorithms can increase the reliability and speed of a search engine. Also, to the best of our knowledge, this is the first time when the ideas of random gradient-free and gradient optimization methods are combined with some efficient method for huge-scale optimization using the concept of an inexact oracle.

**Acknowledgments** The research by P. Dvurechensky and A. Gasnikov presented in Section 4 of this paper was conducted in IITP RAS and supported by the Russian Science Foundation grant (project 14-50-00150), the research presented in Section 5 was supported by RFBR.

## Footnotes

[1] As probablities $[\pi_q^0(\varphi)]_i$, $i \in V_q$, $[P_q(\varphi)]_{\tilde{i},i}$, $\tilde{i} \to i \in E_q$, are scale-invariant ($\pi_q^0(\lambda\varphi) = \pi_q^0(\varphi)$, $P_q(\lambda\varphi) = P_q(\varphi)$), in our experiments, we consider the set $\Phi = \{\varphi \in \mathbb{R}^m : \|\varphi - e_m\|_2 \leq 0.99\}$, where $e_m \in \mathbb{R}^m$ is the vector of all ones, that has large intersection with the simplex $\{\varphi \in \mathbb{R}_{++}^m : \|\varphi\|_1 = 1\}$

[2]yandex.com

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
