[Supplementary Material]

# Learning Supervised PageRank with Gradient-Based and Gradient-Free Optimization Methods

**Lev Bogolubsky**
Yandex
119034 Moscow, Russia
`bogolubsky@yandex-team.ru`

**Pavel Dvurechensky**
Weierstrass Institute
10117 Berlin, Germany;
Institute for Information Transmission Problems RAS
127051 Moscow, Russia
`pavel.dvurechensky@wias-berlin.de`

**Alexander Gasnikov**
Moscow Institute of Physics and Technology
141701, Dolgoprudny, Moscow Region, Russia;
Institute for Information Transmission Problems RAS
127051 Moscow, Russia
`gasnikov@yandex.ru`

**Gleb Gusev**
Yandex
119034 Moscow, Russia
`gleb57@yandex-team.ru`

**Yurii Nesterov**
Center for Operations Research and Econometrics
1348 Louvain-la-Neuve, Belgium;
National Research University Higher School of Economics
125319 Moscow, Russia
`yurii.nesterov@uclouvain.be`

**Andrey Raigorodskii**
Yandex
119034 Moscow, Russia
`raigorodsky@yandex-team.ru`

**Aleksey Tikhonov**
Yandex
119034 Moscow, Russia
`altsoph@yandex-team.ru`

**Maksim Zhukovskii**
Yandex
119034 Moscow, Russia
`zhukmax@yandex-team.ru`

## Abstract

In this paper, we consider a non-convex loss-minimization problem of learning Supervised PageRank models, which can account for some properties not considered by classical approaches such as the classical PageRank model. We propose gradient-based and random gradient-free methods to solve this problem. Our algorithms are based on the concept of an inexact oracle and unlike the state state-of-the-art gradient-based method we manage to provide theoretically the convergence rate guarantees for both of them. In particular, under the assumption of local convexity of the loss function, our random gradient-free algorithm guarantees decrease of the loss function value expectation. At the same time, we theoretically justify that without convexity assumption for the loss function our gradient-based algorithm allows to find a point where the stationary condition is fulfilled with a given accuracy. For

both proposed optimization algorithms, we find the settings of hyperparameters which give the lowest complexity (i.e., the number of arithmetic operations needed to achieve the given accuracy of the solution of the loss-minimization problem). The resulting estimates of the complexity are also provided. Finally, we apply proposed optimization algorithms to the web page ranking problem and compare proposed and state-of-the-art algorithms in terms of the considered loss function.

# 1   INTRODUCTION

The most acknowledged methods of measuring importance of nodes in graphs are based on random walk models. Particularly, PageRank [18], HITS [11], and their variants [8, 9, 19] are originally based on a discrete-time Markov random walk on a link graph. According to the PageRank algorithm, the score of a node equals to its probability in the stationary distribution of a Markov process, which models a random walk on the graph. Despite undeniable advantages of PageRank and its mentioned modifications, these algorithms miss important aspects of the graph that are not described by its structure. In contrast, a number of approaches allows to account for different properties of nodes and edges between them by encoding them in restart and transition probabilities (see [3, 4, 6, 10, 12, 20, 21]). These properties may include, e.g., the statistics about users' interactions with the nodes (in web graphs [12] or graphs of social networks [2]), types of edges (such as URL redirecting in web graphs [20]) or histories of nodes' and edges' changes [22]. Particularly, the transition probabilities in BrowseRank algorithm [12] are proportional to weights of edges which are equal to numbers of users' transitions.

In the general ranking framework called Supervised PageRank [21], weights of nodes and edges in a graph are linear combinations of their features with coefficients as the model parameters. The existing optimization method [21] of learning these parameters and the optimizations methods proposed in the presented paper have two levels. On the lower level, the following problem is solved: to estimate the value of the loss function (in the case of zero-order oracle) and its derivatives (in the case of first-order oracle) for a given parameter vector. On the upper level, the estimations obtained on the lower level of the optimization methods (which we also call inexact oracle information) are used for tuning the parameters by an iterative algorithm. Following [6], the authors of Supervised PageRank consider a non-convex loss-minimization problem for learning the parameters and solve it by a two-level gradient-based method. On the lower level of this algorithm, an estimation of the stationary distribution of the considered Markov random walk is obtained by classical power method and estimations of derivatives w.r.t. the parameters of the random walk are obtained by power method introduced in [23, 24]. On the upper level, the obtained gradient of the stationary distribution is exploited by the gradient descent algorithm. As both power methods give imprecise values of the stationary distribution and its derivatives, there was no proof of the convergence of the state-of-the-art gradient-based method to a local optimum (for locally convex loss functions) or to the stationary point (for not locally convex loss functions).

The considered constrained non-convex loss-minimization problem from [21] can not be solved by existing optimization methods which require exact values of the objective function such as [16] and [7] due to presence of constraints for parameter vector and the impossibility to calculate exact value of the loss function and its gradient. Moreover, standard global optimization methods can not be applied to solve it, because they need access to some stochastic approximation for the loss-function value which in expectation coincides with the true value of the loss-function.

In our paper, we propose two two-level methods to solve the loss-minimization problem from [21]. On the lower level of these methods, we use the linearly convergent method from [17] to calculate an approximation to the stationary distribution of Markov random walk. We analyze other methods from [5] and show that the chosen method is the most suitable since it allows to approximate the value of the loss function with any given accuracy and has lowest complexity estimation among others.

Upper level of the first method is gradient-based. The main obstacle which we have overcome is that the state-of-the-art methods for constrained non-convex optimization assume that the gradient is known exactly, which is not the case in our problem. We develop a gradient method for general constrained non-convex optimization problems with inexact oracle, estimate its convergence rate to the stationary point of the problem. One of the advantages of our method is that it does not require to know the Lipschitz-constant of the gradient of the goal function, which is usually used to define the

stepsize of a gradient algorithm. In order to calculate approximation of the gradient which is used in the upper-level method, we generalize linearly convergent method from [17] (and use it as part of the lower-level method). We prove that it has a linear rate of convergence as well.

Upper level of our second method is random gradient-free. Like for the gradient-based method, we encounter the problem that the existing gradient-free optimization methods [7, 16] require exact values of the objective function. Our contribution to the gradient-free methods framework consists in adapting the approach of [16] to the case of constrained optimization problems when the value of the function is calculated with some known accuracy. We prove a convergence theorem for this method and exploit it on the upper level of the two-level algorithm for solving the problem of learning Supervised PageRank.

Another contribution consists in investigating both for the gradient and gradient-free methods the trade-off between the accuracy of the lower-level algorithm, which is controlled by the number of iterations of method in [17] and its generalization (for derivatives estimation), and the computational complexity of the two-level algorithm as a whole. Finally, we estimate the complexity of the whole two-level algorithms for solving the loss-minimization problem with a given accuracy.

In the experiments, we apply our algorithms to learning Supervised PageRank on real data (we consider the problem of web pages' ranking). We show that both two-level methods outperform the state-of-the-art gradient-based method from [21] in terms of the considered loss function. Summing up, unlike the state-of-the-art method our algorithms have theoretically proven estimates of convergence rate and outperform it in the ranking quality (as we prove experimentally). The main advantages of the first gradient-based algorithm are the following. There is no need to assume that the function is locally convex in order to guarantee that it converges to the stationary point. This algorithm has smaller number of input parameters than gradient-free, because it does not need the Lipschitz constant of the gradient of the loss function. The main advantage of the second gradient-free algorithm is that it avoids calculating the derivative for each element of a large matrix.

The remainder of the paper is organized as follows. In Section 2, we describe the random walk model. In Section 3, we define the loss-minimization problem and discuss its properties. In Section 4, we state two technical lemmas about the numbers of iterations of Nesterov–Nemirovski method (and its generalization) needed to achieve any given accuracy of the loss function (and its gradient). In Section 5 and Section 6 we describe the framework of random gradient-free and gradient-based optimization methods respectively, generalize them to the case when the objective function values and gradients are inaccurate and propose two-level algorithms for the stated loss-minimization problem. Proofs of all our results can be found in Appendix. The experimental results are reported in Section 7. In Section 8, we summarize the outcomes of our study, discuss its benefits and directions of future work.

## 2 MODEL DESCRIPTION

Let $\Gamma = (V, E)$ be a directed graph. As in [21], we suppose that for any $i \in V$ and any $i \to j \in E$, a vector of node's features $\mathbf{V}_i \in \mathbb{R}_+^{m_1}$ and a vector of edge's features $\mathbf{E}_{ij} \in \mathbb{R}_+^{m_2}$ are given. Let $\varphi_1 \in \mathbb{R}^{m_1}$, $\varphi_2 \in \mathbb{R}^{m_2}$ be two vectors of parameters. We denote $m = m_1 + m_2$, $p = |V|$, $\varphi = (\varphi_1, \varphi_2)^T$. Let us describe the random walk on the graph $\Gamma$, which was considered in [21]. A surfer starts a random walk from a random page $i \in U$ ($U$ is some subset in $V$ called *seed set*, $|U| = n$). We assume that $\varphi_1$ and node features are chosen in such way that $\sum_{l \in U} \langle \varphi_1, \mathbf{V}_l \rangle$ is non-zero. The initial probability of being at vertex $i \in V$ is called the *restart probability* and equals

$$[\pi^0(\varphi)]_i = \frac{\langle \varphi_1, \mathbf{V}_i \rangle}{\sum_{l \in U} \langle \varphi_1, \mathbf{V}_l \rangle}, \quad i \in U \tag{2.1}$$

and $[\pi^0(\varphi)]_i = 0$ for $i \in V \setminus U$. At each step, the surfer (with a current position $i \in V$) either chooses with probability $\alpha \in (0, 1)$ (originally [18], $\alpha = 0.15$), which is called the *damping factor*, to go to any vertex from $V$ in accordance with the distribution $\pi^0(\varphi)$ (makes a *restart*) or chooses to traverse an outgoing edge (makes a *transition*) with probability $1 - \alpha$. We assume that $\varphi_2$ and edges features are chosen in such way that $\sum_{l:i \to l} \langle \varphi_2, \mathbf{E}_{il} \rangle$ is non-zero for all $i$ with non-zero outdegree. For $i$ with non-zero outdegree, the probability

$$[P(\varphi)]_{i,j} = \frac{\langle \varphi_2, \mathbf{E}_{ij} \rangle}{\sum_{l:i \to l} \langle \varphi_2, \mathbf{E}_{il} \rangle} \tag{2.2}$$

of traversing an edge $i \rightarrow j \in E$ is called the *transition probability*. If an outdegree of $i$ equals 0, then we set $[P(\varphi)]_{i,j} = [\pi^0(\varphi)]_j$ for all $j \in V$ (the surfer with current position $i$ makes a restart with probability 1). Finally, by Equations 2.1 and 2.2 the total probability of choosing vertex $j \in V$ conditioned by the surfer being at vertex $i$ equals $\alpha[\pi^0(\varphi)]_j + (1 - \alpha)[P(\varphi)]_{i,j}$. Denote by $\pi(\varphi) \in \mathbb{R}^p$ the stationary distribution of the described Markov process. It can be found as a solution of the system of equations

$$\pi = \alpha\pi^0(\varphi) + (1 - \alpha)P^T(\varphi)\pi \tag{2.3}$$

In this paper, we learn the ranking algorithm, which orders the vertices $i$ by their probabilities $[\pi]_i$ in the stationary distribution $\pi$.

## 3  LOSS-MINIMIZATION PROBLEM STATEMENT

Let $Q$ be a set of queries and, for any $q \in Q$, a set of nodes $V_q$ which are relevant to $q$ be given. We are also provided with a ranking algorithm which assigns nodes ranking scores $[\pi_q]_i$, $i \in V_q$, $\pi_q = \pi_q(\varphi)$, as its output. For example, in web search, the score $[\pi_q]_i$ may repesent relevance of the page $i$ w.r.t. the query $q$. Our goal is to find the parameter vector $\varphi$ which minimizes the discrepancy of the ranking scores from the ground truth scoring defined by assessors. For each $q \in Q$, there is a set of nodes in $V_q$ manually judged and grouped by relevance labels $1, \ldots, \ell$. We denote $V_q^j$ the set of nodes annotated with label $\ell + 1 - j$ (i.e., $V_q^1$ is the set of all nodes with the highest relevance score). According to previous studies [12, 21, 22], we consider the square loss function and minimize

$$f(\varphi) = \frac{1}{|Q|} \sum_{q=1}^{|Q|} \|(A_q\pi_q(\varphi))_+\|_2^2 \tag{3.1}$$

as a function of $\varphi$ over some set of feasible values $\Phi$, where vector $x_+$ has components $[x_+]_i = \max\{x_i, 0\}$, the matrices $A_q \in \mathbb{R}^{r_q \times p_q}$, $q \in Q$ represent assessor's view of the relevance of pages to the query $q$, $r_q$ equals $\sum_{1 \leq j < l \leq \ell} |V_q^j||V_q^l|$. We denote $r = \max_{q \in Q} r_q$. By definition each row of matrix $A_q$ corresponds to some pair of pages $i_1 \in V_q^j$, $i_2 \in V_q^l$, where $j < l$, and the $i_1$-th element of this row is equal to $-1$, $i_2$-th element is equal to $1$, and all other elements are equal to $0$.

We consider the ranking algorithm based on scores (2.3) in Markov random walk on a graph $\Gamma_q = (V_q, E_q)$. We assume that feature vectors $\mathbf{V}_i^q$, $i \in V_q$, $\mathbf{E}_{ij}^q$, $i \rightarrow j \in E_q$, depend on $q$ as well. For example, vertices in $V_q$ may represent web pages which were visited by users after submitting a query $q$ and features may reflect different properties of query–page pair. For fixed $q \in Q$, we consider all the objects related to the graph $\Gamma_q$ introduced in the previous section: $U_q := U$, $\pi_q^0 := \pi^0$, $P_q := P$, $p_q := p$, $n_q := n$, $\pi_q := \pi$. This allows ranking model to capture common ("static") dependencies, which do vary between different queries. In this way, the ranking scores depend on query via the "dynamic" (query-dependent) features, but the parameters of the model $\alpha$ and $\varphi$ are not query-dependent. We also denote $p = \max_{q \in Q} p_q$, $n = \max_{q \in Q} n_q$, $s = \max_{q \in Q} s_q$, where $s_q = \max_{i \in V_q} |\{j : i \rightarrow j \in E_q\}|$. In order to guarantee that the probabilities in (2.1) and (2.2) are non-negative and that they do not blow up due to zero value of the denominator, we need appropriately choose the set $\Phi$ of possible values of parameters $\varphi$. Thus we choose some $\hat{\varphi}$ and $R > 0$ such that the set $\Phi$ defined as $\Phi = \{\varphi \in \mathbb{R}^m : \|\varphi - \hat{\varphi}\|_2 \leq R\}$ lies in the set of vectors with positive components $\mathbb{R}_{++}^m$ [1]. The loss-minimization problem which we solve in this paper is as follows

$$\min_{\varphi \in \Phi} f(\varphi), \Phi = \{\varphi \in \mathbb{R}^m : \|\varphi - \hat{\varphi}\|_2 \leq R\}. \tag{3.2}$$

From (2.3), we obtain the following equation for $p_q \times m$ matrix $\frac{d\pi_q(\varphi)}{d\varphi^T}$ which is the derivative of stationary distribution $\pi_q(\varphi)$ with respect to $\varphi$

$$\frac{d\pi_q(\varphi)}{d\varphi^T} = \alpha\frac{d\pi_q^0(\varphi)}{d\varphi^T} + (1 - \alpha)\sum_{i=1}^{p_q} \frac{dp_i(\varphi)}{d\varphi^T}[\pi_q(\varphi)]_i + (1 - \alpha)P_q^T(\varphi)\frac{d\pi_q(\varphi)}{d\varphi^T}, \tag{3.3}$$

where $p_i(\varphi)$ is the $i$-th column of the matrix $P_q^T(\varphi)$. Then the gradient of the function $f(\varphi)$ is easy to derive:

$$\nabla f(\varphi) = \frac{2}{|Q|} \sum_{q=1}^{|Q|} \left( \frac{d\pi_q(\varphi)}{d\varphi^T} \right)^T A_q^T (A_q \pi_q(\varphi))_+. \tag{3.4}$$

# 4 NUMERICAL CALCULATION OF THE VALUE AND THE GRADIENT OF $f(\varphi)$

One of the main difficulties in solving Problem 3.2 is that calculation of the value of the function $f(\varphi)$ requires to calculate $|Q|$ vectors $\pi_q(\varphi)$ which solve (2.3). In our setting, this vector has huge dimension $p_q$ and hence it is computationally very expensive to find it exactly. Moreover, in order to calculate $\nabla f(\varphi)$ one needs to calculate the derivative for each of these huge-dimensional vectors which is also computationally very expensive to be done exactly. At the same time our ultimate goal is to provide methods for solving Problem 3.2 with estimated rate of convergence and complexity. Due to the expensiveness of calculating exact values of $f(\varphi)$ and $\nabla f(\varphi)$ we have to use the framework of optimization methods with inexact oracle which requires to control the accuracy of the oracle, otherwise the convergence is not guaranteed. This means that we need to be able to calculate an approximation to the function $f(\varphi)$ value (inexact zero-order oracle) with a given accuracy for gradient-free methods and approximation to the pair $(f(\varphi), \nabla f(\varphi))$ (inexact first-order oracle) with a given accuracy for gradient methods. Hence we need some numerical scheme which allows to calculate approximation for $\pi_q(\varphi)$ and $\frac{d\pi_q(\varphi)}{d\varphi^T}$ for every $q \in Q$ with a given accuracy.

Motivated by the last requirement we have analysed state-of-the-art methods for finding the solution of Equation 2.3 in huge dimension summarized in the review [5] and power method, used in [18, 2, 21]. Only four methods allow to make the difference $\|\pi_q(\varphi) - \tilde{\pi}_q\|$, where $\tilde{\pi}_q$ is the approximation, small for some norm $\| \cdot \|$ which is crucial to estimate the error in the approximation of the function $f(\varphi)$ value. These methods are: Markov Chain Monte Carlo (MCMC), Spillman's, Nesterov-Nemirovski's (NN) [17] and power method. Spillman's algoritm and power method converge in infinity norm which is usually $p_q$ times larger than 1-norm. MCMC converges in 2-norm which is usually $\sqrt{p_q}$ times larger than 1-norm. Also MCMC is randomized and converges only in average which makes it hard to control the accuracy of the approximation $\tilde{\pi}_q$. Unlike the other three, NN is deterministic and converges in 1-norm which gives minimum $\sqrt{p_q}$ times better approximation. At the same time, to the best of our knowledge, NN method is the only method that admits a generalization which, as we prove in this paper, calculates the derivative $\frac{d\pi_q(\varphi)}{d\varphi^T}$ with any given accuracy.

The method [17] for approximation of $\pi_q(\varphi)$ for any fixed $q \in Q$ constructs a sequence $\pi_k$ and the output $\tilde{\pi}_q(\varphi, N)$ (for some fixed non-negative integer $N$) by the following rule

$$\pi_0 = \pi_q^0(\varphi), \quad \pi_{k+1} = P_q^T(\varphi)\pi_k, \quad \tilde{\pi}_q(\varphi, N) = \frac{\alpha}{1 - (1-\alpha)^{N+1}} \sum_{k=0}^{N} (1-\alpha)^k \pi_k. \tag{4.1}$$

**Lemma 1.** *Assume that for some $\delta_1 > 0$ Method 4.1 with $N = \left\lceil \frac{1}{\alpha} \ln \frac{8r}{\delta_1} \right\rceil - 1$ is used to calculate the vector $\tilde{\pi}_q(\varphi, N)$ for every $q \in Q$. Then*

$$\tilde{f}(\varphi, \delta_1) = \frac{1}{|Q|} \sum_{q=1}^{|Q|} \|(A_q \tilde{\pi}_q(\varphi, N))_+\|_2^2 \tag{4.2}$$

*satisfies*

$$|\tilde{f}(\varphi, \delta_1) - f(\varphi)| \leq \delta_1. \tag{4.3}$$

*Moreover, the calculation of $\tilde{f}(\varphi, \delta_1)$ requires not more than $|Q|(3mps + 3psN + 6r)$ a.o. and not more than $3ps$ memory items.*

The proof of Lemma 1 can be found in Appendix A.1.

Our generalization of the method [17] for calculation of $\frac{d\pi_q(\varphi)}{d\varphi^T}$ for any $q \in Q$ is the following. Choose some non-negative integer $N_1$ and calculate $\tilde{\pi}_q(\varphi, N_1)$ using (4.1). Calculate a sequence $\Pi_k$

$$\Pi_0 = \alpha \frac{d\pi_q^0(\varphi)}{d\varphi^T} + (1 - \alpha) \sum_{i=1}^{p_q} \frac{dp_i(\varphi)}{d\varphi^T} [\tilde{\pi}_q(\varphi, N_1)]_i, \quad \Pi_{k+1} = P_q^T(\varphi)\Pi_k. \quad (4.4)$$

The output is (for some fixed non-negative integer $N_2$)

$$\tilde{\Pi}_q(\varphi, N_2) = \frac{1}{1 - (1-\alpha)^{N_2+1}} \sum_{k=0}^{N_2} (1-\alpha)^k \Pi_k. \quad (4.5)$$

In what follows, we use the following norm on the space of matrices $A \in \mathbb{R}^{n_1 \times n_2}$: $\|A\|_1 = \max_{j=1,...,n_2} \sum_{i=1}^{n_1} |a_{ij}|$.

**Lemma 2.** *Let $\beta_1$ be a number (explicitly computable, see Appendix A.2 Equation A.13) such that for all $\varphi \in \Phi$*

$$\alpha \left\| \frac{d\pi_q^0(\varphi)}{d\varphi^T} \right\|_1 + (1-\alpha) \sum_{i=1}^{p_q} \left\| \frac{dp_i(\varphi)}{d\varphi^T} \right\|_1 \le \beta_1. \quad (4.6)$$

*Assume that Method 4.1 with $N_1 = \left\lceil \frac{1}{\alpha} \ln \frac{24\beta_1 r}{\alpha \delta_2} \right\rceil - 1$ is used for every $q \in Q$ to calculate the vector $\tilde{\pi}_q(\varphi, N_1)$ and Method 4.4, 4.5 with $N_2 = \left\lceil \frac{1}{\alpha} \ln \frac{8\beta_1 r}{\alpha \delta_2} \right\rceil - 1$ is used for every $q \in Q$ to calculate the matrix $\tilde{\Pi}_q(\varphi, N_2)$ (4.5). Then the vector*

$$\tilde{g}(\varphi, \delta_2) = \frac{2}{|Q|} \sum_{q=1}^{|Q|} \left( \tilde{\Pi}_q(\varphi, N_2) \right)^T A_q^T (A_q \tilde{\pi}_q(\varphi, N_1))_+ \quad (4.7)$$

*satisfies*

$$\|\tilde{g}(\varphi, \delta_2) - \nabla f(\varphi)\|_\infty \le \delta_2. \quad (4.8)$$

*Moreover the calculation of $\tilde{g}(\varphi, \delta_2)$ requires not more than $|Q|(10mps + 3psN_1 + 3mpsN_2 + 7r)$ a.o. and not more than $4ps + 4mp + r$ memory items.*

The proof of Lemma 2 can be found in Appendix A.2.

## 5 RANDOM GRADIENT-FREE OPTIMIZATION METHODS

In this section, we first describe general framework of random gradient-free methods with inexact oracle and then apply it for Problem 3.2. Lemma 1 allows to control the accuracy of the inexact zero-order oracle and hence apply random gradient-free methods with inexact oracle.

### 5.1 GENERAL FRAMEWORK

Below we extend the framework of random gradient-free methods [1, 16, 7] for the situation of presence of uniformly bounded error of unknown nature in the value of an objective function in general optimization problem. Unlike [16], we consider a constrained optimization problem and a randomization on a Euclidean sphere which seems to give better large deviations bounds and doesn't need the assumption that the objective function can be calculated at any point of $\mathbb{R}^m$.

Let $\mathcal{E}$ be a $m$-dimensional vector space. In this subsection, we consider a general function $f(\cdot) : \mathcal{E} \to \mathbb{R}$ and denote its argument by $x$ or $y$ to avoid confusion with other sections. We denote the value of linear function $g \in \mathcal{E}^*$ at $x \in \mathcal{E}$ by $\langle g, x \rangle$. We choose some norm $\|\cdot\|$ in $\mathcal{E}$ and say that $f \in C_L^{1,1}(\|\cdot\|)$ iff

$$|f(x) - f(y) - \langle \nabla f(y), x - y \rangle| \le \frac{L}{2} \|x - y\|^2, \quad \forall x, y \in \mathcal{E}. \quad (5.1)$$

The problem of our interest is to find $\min_{x \in X} f(x)$, where $f \in C_L^{1,1}(\|\cdot\|)$, $X$ is a closed convex set and there exists a number $D \in (0, +\infty)$ such that $\text{diam} X := \max_{x,y \in X} \|x - y\| \le D$. Also

we assume that the inexact zero-order oracle for $f(x)$ returns a value $\tilde{f}(x, \delta) = f(x) + \tilde{\delta}(x)$, where $\tilde{\delta}(x)$ is the error satisfying for some $\delta > 0$ (which is known) $|\tilde{\delta}(x)| \leq \delta$ for all $x \in X$. Let $x^* \in \arg\min_{x \in X} f(x)$. Denote $f^* = \min_{x \in X} f(x)$.

Unlike [16], we define the biased gradient-free oracle $g_\tau(x, \delta) = \frac{m}{\tau}(\tilde{f}(x + \tau\xi, \delta) - \tilde{f}(x, \delta))\xi$, where $\xi$ is a random vector uniformly distributed over the unit sphere $\mathcal{S} = \{t \in \mathbb{R}^m : \|t\|_2 = 1\}$, $\tau$ is a smoothing parameter.

Algorithm 1 below is the variation of the projected gradient descent method. Here $\Pi_X(x)$ denotes the Euclidean projection of a point $x$ onto the set $X$.

---

**Algorithm 1** Gradient-type method

    **Input:** Point $x_0 \in X$, stepsize $h > 0$, number of steps $M$.
    Set $k = 0$.
    **repeat**
        Generate $\xi_k$ and calculate corresponding $g_\tau(x_k, \delta)$.
        Calculate $x_{k+1} = \Pi_X(x_k - hg_\tau(x_k, \delta))$.
        Set $k = k + 1$.
    **until** $k > M$
    **Output:** The point $y_M = \arg\min_x\{f(x) : x \in \{x_0, \dots, x_M\}\}$.

---

Next theorem gives the convergence rate of Algorithm 1. Denote by $\Xi_k = (\xi_0, \dots, \xi_k)$ the history of realizations of the vector $\xi$ generated on each iteration of the algorithm.

**Theorem 1.** *Let $f \in C_L^{1,1}(\|\cdot\|_2)$ and convex. Assume that $x^* \in \mathrm{int}X$, and the sequence $x_k$ is generated by Algorithm 1 with $h = \frac{1}{8mL}$. Then for any $M \geq 0$, we have*

$$\mathbb{E}_{\Xi_{M-1}} f(y_M) - f^* \leq \frac{8mLD^2}{M + 1} + \frac{\tau^2 L(m + 8)}{8} + \frac{\delta mD}{4\tau} + \frac{\delta^2 m}{L\tau^2}. \tag{5.2}$$

The full proof of the theorem is in Appendix B.

It is easy to see that to make the right hand side of (5.2) less than a desired accuracy $\varepsilon$ it is sufficient to choose

$$M = \left\lceil \frac{32mLD^2}{\varepsilon} \right\rceil, \quad \tau = \sqrt{\frac{2\varepsilon}{L(m + 8)}}, \quad \delta \leq \frac{\varepsilon^{\frac{3}{2}}\sqrt{2}}{8mD\sqrt{L(m + 8)}}. \tag{5.3}$$

## 5.2 SOLVING THE LEARNING PROBLEM

In this subsection, we apply the results of the previous subsection to solve Problem 3.2 in the following way. Note that we can not directly apply the results of [16] due to presence of constraints and inexactness of the oracle. We assume that the set $\Phi$ is a small vicinity of some local minimum $\varphi^*$ and the function $f(\varphi)$ is convex in this vicinity (generally speaking, the function defined in (3.1) is nonconvex). We choose the desired accuracy $\varepsilon$ for approximation of the optimal value $f^*$ in this problem. This accuracy in accordance with (5.3) gives us the number of steps of Algorithm 1, the value of the parameter $\tau$, the value of the required accuracy $\delta$ of the inexact zero-order oracle. Knowing the value $\delta$, using Lemma 1 we choose the number of steps $N$ of Method 4.1 and calculate an approximation $\tilde{f}(\varphi, \delta)$ for the function $f(\varphi)$ value with accuracy $\delta$. Then we use the inexact zero-order oracle $\tilde{f}(\varphi, \delta)$ to make a step of Algorithm 1. Theorem 1 and the fact that the feasible set $\Phi$ is a Euclidean ball makes it natural to choose $\|\cdot\|_2$-norm in the space $\mathbb{R}^m$ of parameter $\varphi$. It is easy to see that in this norm $\mathrm{diam}\Phi \leq 2R$. Algorithm 2 is a formal record of these ideas. To the best of our knowledge, this is the first time when the idea of random gradient-free optimization methods is combined with some efficient method for huge-scale optimization using the concept of an inexact zero-order oracle.

The most computationally hard on each iteration of the main cycle of this method are calculations of $\tilde{f}(\varphi_k + \tau\xi_k, \delta)$, $\tilde{f}(\varphi_k, \delta)$. Using Lemma 1, we obtain that each iteration of Algorithm 2 needs not

---

**Algorithm 2** Gradient-free method for Problem 3.2

---

**Input:** Point $\varphi_0 \in \Phi$, $L$ – Lipschitz constant for the function $f(\varphi)$ on $\Phi$, accuracy $\varepsilon > 0$.

Define $M = \left\lceil 128m \frac{LR^2}{\varepsilon} \right\rceil$, $\delta = \frac{\varepsilon^{\frac{3}{2}} \sqrt{2}}{16mR\sqrt{L(m+8)}}$, $\tau = \sqrt{\frac{2\varepsilon}{L(m+8)}}$.

Set $k = 0$.

**repeat**

    Generate random vector $\xi_k$ uniformly distributed over a unit Euclidean sphere $\mathcal{S}$ in $R^m$.

    Calculate $\tilde{f}(\varphi_k + \tau\xi_k, \delta)$, $\tilde{f}(\varphi_k, \delta)$ using Lemma 1 with $\delta_1 = \delta$.

    Calculate $g_\tau(\varphi_k, \delta) = \frac{m}{\tau}(\tilde{f}(\varphi_k + \tau\xi_k, \delta) - \tilde{f}(\varphi_k, \delta))\xi_k$.

    Calculate $\varphi_{k+1} = \Pi_\Phi\left(\varphi_k - \frac{1}{8mL}g_\tau(\varphi_k, \delta)\right)$.

    Set $k = k + 1$.

**until** $k > M$

**Output:** The point $\hat{\varphi}_M = \arg\min_\varphi\{f(\varphi) : \varphi \in \{\varphi_0, \ldots, \varphi_M\}\}$.

---

more than

$$2|Q|\left(3mps + \frac{3ps}{\alpha}\ln\frac{128mrR\sqrt{L(m+8)}}{\varepsilon^{3/2}\sqrt{2}} + 6r\right)$$

a.o. So, we obtain the following result, which gives the complexity of Algorithm 2.

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

*Moreover, the total number of checks of Inequality 6.4 is not more than $M + \log_2 \frac{2L}{L_0}$.*

The full proof of the theorem is in Appendix C.

It is easy to show that when $\|M_K(x_K - x_{K+1})\|^2 \leq \varepsilon$ for small $\varepsilon$, then for all $x \in X$ it holds that $\langle \nabla f(x_{K+1}) + \nabla h(x_{K+1}), x - x_{K+1} \rangle \geq -c\sqrt{\varepsilon}$, where $c > 0$ is a constant, $\nabla h(x_{K+1})$ is some subgradient of $h(x)$ at $x_{K+1}$. This means that at the point $x_{K+1}$ the necessary condition of a local minimum is fulfilled with a good accuracy, i.e. $x_{K+1}$ is a good approximation of a stationary point.

## 6.2 SOLVING THE LEARNING PROBLEM

In this subsection, we return to Problem 3.2 and apply the results of the previous subsection. Note that we can not directly apply the results of [7] due inexactness of the oracle. For this problem, $h(\cdot) \equiv 0$. It is easy to show that in 1-norm $\operatorname{diam}\Phi \leq 2R\sqrt{m}$. For any $\delta > 0$, Lemma 1 with $\delta_1 = \frac{\delta}{2}$ allows us to obtain $\tilde{f}(\varphi, \delta_1)$ such that Inequality 4.3 holds and Lemma 2 with $\delta_2 = \frac{\delta}{4R\sqrt{m}}$ allows us to obtain $\tilde{g}(\varphi, \delta_2)$ such that Inequality 4.8 holds. Similar to [25], since $f \in C_L^{1,1}(\| \cdot \|_2)$, these two inequalities lead to Inequality 6.2 for $\tilde{f}(\varphi, \delta_1)$ in the role of $\tilde{f}(x, \delta)$, $\tilde{g}(\varphi, \delta_2)$ in the role of $\tilde{g}(x, \delta)$ and $\| \cdot \|_2$ in the role of $\| \cdot \|$.

We choose the desired accuracy $\varepsilon$ for approximating the stationary point of Problem 3.2. This accuracy gives the required accuracy $\delta$ of the inexact first-order oracle for $f(\varphi)$ on each step of the inner cycle of the Algorithm 3. Knowing the value $\delta_1 = \frac{\delta}{2}$ and using Lemma 1, we choose the number of steps $N$ of Algorithm 4.1 and thus approximate $\tilde{f}(\varphi)$ with the required accuracy $\delta_1$ by

$\tilde{f}(\varphi, \delta_1)$. Knowing the value $\delta_2 = \frac{\delta}{4R\sqrt{m}}$ and using Lemma 2, we choose the number of steps $N_1$ of Method 4.1 and the number of steps $N_2$ of Method 4.4, 4.5 and obtain the approximation $\tilde{g}(\varphi, \delta_2)$ of $\nabla f(\varphi)$ with the required accuracy $\delta_2$. Then we use the inexact first-order oracle $(\tilde{f}(\varphi, \delta_1), \tilde{g}(\varphi, \delta_2))$ to perform a step of Algorithm 3.

Since $\Phi$ is the Euclidean ball, it is natural to set $\mathcal{E} = R^m$ and $\|\cdot\| = \|\cdot\|_2$, choose the prox-function $d(\varphi) = \frac{1}{2}\|\varphi\|_2^2$. Then the Bregman distance is $V(\varphi, \omega) = \frac{1}{2}\|\varphi - \omega\|_2^2$.

Algorithm 4 is a formal record of the above ideas. To the best of our knowledge, this is the first time when the idea of gradient optimization methods is combined with some efficient method for huge-scale optimization using the concept of an inexact first-order oracle.

---

**Algorithm 4** Adaptive gradient method for Problem 3.2

**Input:** Point $\varphi_0 \in \Phi$, number $L_0 > 0$, accuracy $\varepsilon > 0$.
Set $k = 0$, $z = +\infty$.
**repeat**
  Set $M_k = L_k$, flag $= 0$.
  **repeat**
    Set $\delta_1 = \frac{\varepsilon}{32M_k}$, $\delta_2 = \frac{\varepsilon}{64M_k R\sqrt{m}}$.
    Calculate $\tilde{f}(\varphi_k, \delta_1)$ using Lemma 1 and $\tilde{g}(\varphi_k, \delta_2)$ using Lemma 2.
    Find
$$\omega_k = \arg\min_{\varphi \in \Phi}\left\{\langle \tilde{g}(\varphi_k, \delta_2), \varphi\rangle + \frac{M_k}{2}\|\varphi - \varphi_k\|_2^2.\right\}$$
    Calculate $\tilde{f}(\omega_k, \delta_1)$ using Lemma 1.
    If the inequality
$$\tilde{f}(\omega_k, \delta_1) \le \tilde{f}(\varphi_k, \delta_1) + \langle \tilde{g}(\varphi_k, \delta_2), \omega_k - \varphi_k\rangle + \frac{M_k}{2}\|\omega_k - \varphi_k\|_2^2 + \frac{\varepsilon}{8M_k}$$
    holds, set flag $= 1$. Otherwise set $M_k = 2M_k$.
  **until** flag $= 1$
  Set $\varphi_{k+1} = \omega_k$, $L_{k+1} = \frac{M_k}{2}$, .
  If $\|M_k(\varphi_k - \varphi_{k+1})\|_2 < z$, set $z = \|M_k(\varphi_k - \varphi_{k+1})\|_2$, $K = k$.
  Set $k = k + 1$.
**until** $z \le \varepsilon$
**Output:** The point $\varphi_{K+1}$.

---

The most computationally consuming operations of the inner cycle of Algorithm 4 are calculations of $\tilde{f}(\varphi_k, \delta_1)$, $\tilde{f}(\omega_k, \delta_1)$ and $\tilde{g}(\varphi_k, \delta_2)$. Using Lemma 1 and Lemma 2, we obtain that each inner iteration of Algorithm 4 needs not more than
$$7r|Q| + \frac{6mps|Q|}{\alpha}\ln\frac{1024\beta_1 rRL\sqrt{m}}{\alpha\varepsilon}$$
a.o. Using Theorem 3, we obtain the following result, which gives the complexity of Algorithm 4.

**Theorem 4.** *The total number of arithmetic operations in Algorithm 4 for the accuracy $\varepsilon$ (i.e. for the inequality $\|M_K(\varphi_K - \varphi_{K+1})\|_2^2 \le \varepsilon$ to hold) is not more than*
$$\left(\frac{8L(f(\varphi_0) - f^*)}{\varepsilon} + \log_2\frac{2L}{L_0}\right) \cdot \left(7r|Q| + \frac{6mps|Q|}{\alpha}\ln\frac{1024\beta_1 rRL\sqrt{m}}{\alpha\varepsilon}\right).$$

## 7 EXPERIMENTAL RESULTS

We apply different learning techniques, our gradient-free and gradient-based methods and state-of-the-art gradient-based method, to the web page ranking problem and compare their performances. In the next section, we describe the graph and the dataset, which we exploit in our experiments. In Section 7.2, we describe the results of the experiments.

## 7.1 DATA

In our experiments, we consider the user web browsing graph $\Gamma_q = (V_q, E_q)$, $q \in Q$ (which was first considered in [12]). We choose the user browsing graph instead of a link graph with the purpose to make the model query-dependent. In this graph, the set of vertices consists of all the distinct elements from all the sessions which are started from $q$. The set of directed edges $E_q$ represents all the ordered pairs of neighboring elements $(\tilde{i}, i)$ from such sessions. We add a page $i$ in the seed set $U_q$ if and only if there is a session which is started from $q$ and contains $i$ as its first element.

All experiments are performed with pages and links crawled by a popular commercial search engine. We randomly choose the set of queries $Q$ the user sessions start from, which contains 600 queries. There are $\approx 11.7K$ vertices and $\approx 7.5K$ edges in graphs $\Gamma_q$, $q \in Q$, in total. For each query, a set of pages was judged by professional assessors hired by the search engine. Our data contains $\approx 1.7K$ judged query–document pairs. The relevance score is selected from among 5 labels. We divide our data into two parts. On the first part $Q_1$ (50% of the set of queries $Q$) we train the parameters and on the second part $Q_2$ we test the algorithms. To define weights of nodes and edges we consider a set of $m_1 = 26$ query–document features. For any $q \in Q$ and $i \in V_q$, the vector $\mathbf{V}_i^q$ contains values of all these features for query–document pair $(q, i)$. The vector of $m_2 = 52$ features $\mathbf{E}_{\tilde{i}i}^q$ for an edge $\tilde{i} \to i \in E_q$ is obtained simply by concatenation of the feature vectors of pages $\tilde{i}$ and $i$.

To study a dependency between the efficiency of the algorithms and the sizes of the graphs, we sort the sets $Q_1, Q_2$ in ascending order of sizes of the respective graphs. Sets $Q_j^1, Q_j^2, Q_j^3$ contain first (in terms of these order) $100, 200, 300$ elements respectively for $j \in \{1, 2\}$.

## 7.2 PERFORMANCES OF THE OPTIMIZATION ALGORITHMS

We find the optimal values of the parameters $\varphi$ by all the considered methods (our gradient-free method GFN (Algorithm 2), the gradient-based method GBN (Algorithm 4), the state-of-the-art gradient-method GBP), which solve Problem 3.1.

The sets of hyperparameters which are exploited by the optimization methods (and not tuned by them) are the following: the Lipschitz constant $L = 10^{-4}$ in GFN (and $L_0 = 10^{-4}$ in GBN), the accuracy $\varepsilon = 10^{-6}$ (in both GBN and GFN), the radius $R = 0.99$ (in both GBN and GFN). On all sets of queries, we compare final values of the loss function for GBN when $L_0 \in \{10^{-4}, 10^{-3}, 10^{-2}, 10^{-1}, 1\}$. The differences are less than $10^{-7}$. We choose $L$ in GFN to be equal to $L_0$. On Figure 2, we show how the choice of $L$ influences the output of the gradient-free algorithm. Moreover, we evaluate both our gradient-based and gradient-free algorithms for different values of the accuracies. The outputs of the algorithms differ insufficiently on all test sets $Q_2^i$, $i \in \{1, 2, 3\}$, when $\varepsilon \leq 10^{-6}$. On the lower level of the state-of-the-art gradient-based algorithm, the stochastic matrix and its derivative are raised to the powers $N_1$ and $N_2$ respectively. We choose $N_1 = N_2 = 100$, since the outputs of the algorithm differ insufficiently on all test sets, when $N_1 \geq 100$, $N_2 \geq 100$. We evaluate GBP for different values of the step size $(50, 100, 200, 500)$. We stop the GBP algorithms when the differences between the values of the loss function on the next step and the current step are less than $-10^{-5}$ on the test sets. On Figure 1, we give the outputs of the optimization algorithms on each iteration of the upper levels of the learning processes on the test sets.

In Table 1, we present the performances of the optimization algorithms in terms of the loss function $f$ (3.1). We also compare the algorithms with the untuned Supervised PageRank ($\varphi = \varphi_0 = e_m$).

GFN significantly outperforms the state-of-the-art algorithms on all test sets. GBN significantly outperforms the state-of-the-art algorithm on $Q_2^1$ (we obtain the $p$-values of the paired $t$-tests for all the above differences on the test sets of queries, all these values are less than 0.005). However, GBN requires less iterations of the upper level (until it stops) than GBP for step sizes 50 and 100 on $Q_2^2, Q_2^3$.

Finally, we show that Nesterov–Nemirovski method converges to the stationary distribution faster than the power method. On Figure 2, we demonstrate the dependencies of the value of the loss function on $Q_1^1$ for both methods of computing the untuned Supervised PageRank ($\varphi = \varphi_0 = e_m$).

Figure 1: Values of the loss function on each iteration of the optimization algorithms on the test sets.

## 8   DISCUSSIONS AND CONCLUSIONS

Let us note that Theorem 1 allows to estimate the probability of large deviations using the obtained mean rate of convergence for Algorithm 1 (and hence Algorithm 2) in the following way. If $f(x)$ is $\mu$-strongly convex, then we prove (see Appendix) a geometric mean rate of convergence: $\mathbb{E}_{\Xi_{M-1}} f(x_M) - f^* \leq O\left(m\frac{L}{\mu}\ln\left(\frac{LD^2}{\varepsilon}\right)\right)$. Using Markov's inequality, we obtain that after $O\left(m\frac{L}{\mu}\ln\left(\frac{LD^2}{\varepsilon\sigma}\right)\right)$ iterations the inequality $f(x_M) - f^* \leq \varepsilon$ holds with a probability greater than $1 - \sigma$, where $\sigma \in (0, 1)$ is a desired confidence level. If the function $f(x)$ is convex, but not strongly convex, then we can introduce the regularization with the parameter $\mu = \varepsilon/D^2$ minimizing the function $f(x) + \frac{\mu}{2}\|x - \hat{x}\|_2^2$ ($\hat{x}$ is some point in the set $X$), which is strongly convex. This will give us that after $O\left(m\frac{LD^2}{\varepsilon}\ln\left(\frac{LD^2}{\varepsilon\sigma}\right)\right)$ iterations the inequlity $f(x_M) - f^* \leq \varepsilon$ holds with a probability greater than $1 - \sigma$.

| Meth. | $Q_2^1$ | | $Q_2^2$ | | $Q_2^3$ | |
|---|---|---|---|---|---|---|
| | loss | steps | loss | steps | loss | steps |
| PR | .00357 | 0 | .00354 | 0 | .0033 | 0 |
| GBN | .00279 | 12 | .00305 | 12 | .00295 | 12 |
| GFN | .00274 | $10^6$ | .00297 | $10^6$ | .00292 | $10^6$ |
| GBP 50s. | .00282 | 16 | .00307 | 31 | .00295 | 40 |
| GBP 100s. | .00282 | 8 | .00307 | 16 | .00295 | 20 |
| GBP 200s. | .00283 | 4 | .00308 | 7 | .00295 | 9 |
| GBP 500s. | .00283 | 2 | .00308 | 2 | .00295 | 3 |

Table 1: Comparison of the algorithms on the test sets.

Figure 2: Comparison of convergence rates of the power method and the method of Nesterov and Nemirovski (on the left) & loss function values on each iteration of GFN with different values of the parameter $L$ on the train set $Q_1^1$

We consider a problem of learning parameters of Supervised PageRank models, which are based on calculating the stationary distributions of the Markov random walks with transition probabilities depending on the parameters. Due to the impossibility of exact calculating derivatives of the stationary distributions w.r.t. its parameters, we propose two two-level loss-minimization methods with inexact oracle to solve it instead of the previous gradient-based approach. For both proposed optimization algorithms, we find the settings of hyperparameters which give the lowest complexity (i.e., the number of arithmetic operations needed to achieve the given accuracy of the solution of the loss-minimization problem).

We apply our algorithm to the web page ranking problem by considering a dicrete-time Markov random walk on the user browsing graph. Our experiments show that our gradient-free method outperforms the state-of-the-art gradient-based method. For one of the considered test sets, our gradient-based method outperforms the state-of-the-art as well. For other test sets, the differences in the values of the loss function are insignificant. Moreover, we prove that under the assumption of local convexity of the loss function, our random gradient-free algorithm guarantees decrease of the loss function value expectation. At the same time, we theoretically justify that without convexity assumption for the loss function our gradient-based algorithm allows to find a point where the stationary condition is fulfilled with a given accuracy.

In future, it would be interesting to apply our algorithms to other ranking problems.

**Acknowledgments** The research by P. Dvurechensky and A. Gasnikov presented in Section 5 of this paper was conducted in IITP RAS and supported by the Russian Science Foundation grant (project 14-50-00150), the research presented in Section 6 was supported by RFBR.

## Footnotes

[1] As probablities $[\pi_q^0(\varphi)]_i$, $i \in V_q$, $[P_q(\varphi)]_{\tilde{i},i}$, $\tilde{i} \rightarrow i \in E_q$, are scale-invariant ($\pi_q^0(\lambda\varphi) = \pi_q^0(\varphi)$, $P_q(\lambda\varphi) = P_q(\varphi)$), in our experiments, we consider the set $\Phi = \{\varphi \in \mathbb{R}^m : \|\varphi - e_m\|_2 \leq 0.99\}$, where $e_m \in \mathbb{R}^m$ is the vector of all ones, that has large intersection with the simplex $\{\varphi \in \mathbb{R}_{++}^m : \|\varphi\|_1 = 1\}$

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

# A    Missed proofs for Section 4

## A.1    Proof of Lemma 1

Let for any $x \in \mathbb{R}^p$ $\|x\|_1 = \sum_{i=1}^{p} |x_i|$ be its 1-norm, $\|x\|_2 = \sqrt{\sum_{i=1}^{p} |x_i|^2}$ be its standard Euclidean norm and $\|x\|_\infty = \max_{i=1,\ldots,p} |x_i|$ be its max-norm.

First in Lemma A.1 we estimate the complexity of Method 4.1 in terms of the number of iterations and number of arythmetic operations which are required to approximate the solution of Equation (2.3) with a given accuracy. Then we prove technical Lemma A.2 which is used in the proof of Lemma A.3, which tells how the error of the approximate solution Equation (2.3) affects the error in the funcation $f(\varphi)$ value. Finally we combine Lemma A.1 and Lemma A.3 to prove Lemma 1.

**Lemma A.1.** *Let us fix some $q \in Q$. Let $\pi_q^0(\varphi)$ be defined in (2.1), matrices $P_q(\varphi)$ be defined in (2.2). Assume that Method 4.1 with*

$$N = \left\lceil \frac{1}{\alpha} \ln \frac{2}{\Delta_1} \right\rceil - 1$$

*is used to calculate the approximation $\tilde{\pi}_q(\varphi, N)$ to the ranking vector $\pi_q(\varphi)$ which is the solution of Equation 2.3. Then the vector $\tilde{\pi}_q(\varphi, N)$ satisfies*

$$\|\tilde{\pi}_q(\varphi, N) - \pi_q(\varphi)\|_1 \leq \Delta_1 \tag{A.1}$$

*and its calculation requires not more than*

$$3mp_q s_q + 3p_q s_q N$$

*a.o. and not more than*

$$2p_q s_q$$

*memory amount additionally to the memory which is needed to store all the data about features and matrices $A_q, b_q, q \in Q$.*

**Proof.** As it is shown in [17] the vector $\tilde{\pi}_q(\varphi, N)$ satisfies

$$\|\tilde{\pi}_q(\varphi, N) - \pi_q(\varphi)\|_1 \leq 2(1 - \alpha)^{N+1}. \tag{A.2}$$

Since for any $\alpha \in (0, 1]$ it holds that $\alpha \leq \ln \frac{1}{1-\alpha}$ we have from the lemma assumption that

$$N + 1 \geq \frac{1}{\alpha} \ln \frac{2}{\Delta_1} \geq \frac{\ln \frac{2}{\Delta_1}}{\ln \frac{1}{1-\alpha}}.$$

This gives us that $2(1 - \alpha)^{N+1} \leq \Delta_1$ which in combination with (A.2) gives (A.1).

Let us estimate the number of a.o and the memory amount used for calculations. We will go through Method 4.1 step by step and estimate from above the number of a.o. for each step. Since we need to estimate from above the total number of a.o. used for the whole algorithm we will update this upper bound (and denote it by TAO) by adding on each step the obtained upper bound of a.o. number for this step. On each step we also estimate from above (and denote this estimate by MM) maximum memory amount which was used by Method 4.1 before the end of this step. Finally, at the end of each step we estimate from above by UM the memory amount which is still occupied besides the step is finished.

1. First iteration of this method requires to calculate $\pi = \pi_q^0$. The variable $\pi$ will store current (in terms of steps in $k$) iterate $\pi_k$ which potentially has $p_q$ non-zero elements. In accordance to its definition (2.1) one has for all $i \in U_q$

$$[\pi_q^0]_i = \frac{\langle \varphi_1, \mathbf{V}_i^q \rangle}{\sum_{j \in U_q} \langle \varphi_1, \mathbf{V}_j^q \rangle}$$

   (a) We calculate $\langle \varphi_1, \mathbf{V}_i^q \rangle$ for all $i \in U_q$ and store the result. This requires $2m_1 n_q$ a.o. and not more than $p_q$ memory items since $|U_q| = n_q \leq p_q$ and $\mathbf{V}_j^q \in \mathbb{R}^{m_1}$ for all $i \in U_q$.

   (b) We calculate $\frac{1}{\sum_{j \in U_q} \langle \varphi_1, \mathbf{V}_j^q \rangle}$ which requires $n_q$ a.o. and 2 memory items.

   (c) We calculate $\frac{\langle \varphi_1, \mathbf{V}_i^q \rangle}{\sum_{j \in U_q} \langle \varphi_1, \mathbf{V}_j^q \rangle}$ for all $i \in U_q$. This needs $n_q$ a.o. and no additional memory.

   So after this stage MM $= p_q + 2$, UM $= p_q$, TAO $= 2m_1 n_q + 2n_q$.

2. We need to calculate elements of matrix $P_q(\varphi)$. In accordance to (2.2) one has

$$[P_q(\varphi)]_{ij} = \frac{\langle \varphi_2, \mathbf{E}_{ij}^q \rangle}{\sum_{l:i \to l} \langle \varphi_2, \mathbf{E}_{il}^q \rangle}.$$

   This means that one needs to calculate $p_q$ vectors like $\pi_q^0$ on the previous step but each with not more than $s_q$ non-zero elements and dimension of $\varphi_2$ equal to $m_2$. Thus we need $p_q(2m_2 s_q + 2s_q)$ a.o. and not more than $p_q s_q + 2$ memory items additionally to $p_q$ memory items already used. At the end of this stage we have TAO $= 2m_1 n_q + 2n_q + p_q(2m_2 s_q + 2s_q)$, MM $= p_q + 2 + p_q s_q$ and UM $= p_q + p_q s_q$ since we store $\pi$ and $P_q(\varphi)$ in memory.

3. We set $\tilde{\pi}_q^N = \pi_q^0$ (this variable will store current approximation of $\tilde{\pi}_q^N$ which potentially has $p_q$ non-zero elements). This requires $n_q$ a.o. and $p_q$ memory items. Also we set $a = (1 - \alpha)$. At the end of this step we have TAO $= 2m_1 n_q + 2n_q + p_q(2m_2 s_q + 2s_q) + n_q + 1$, MM $= p_q + 2 + p_q s_q + p_q$ and UM $= p_q + p_q s_q + p_q + 1$.

4. For every step from 1 to $N$

    (a) We set $\pi_1 = P_q^T(\varphi)\pi$. This requires not more than $2p_q s_q$ a.o. since the number of non-zero elements in the matrix $P_q^T(\varphi)$ is not more than $p_q s_q$ and we need to multiply each element by some element of $\pi$ and add it to the sum. Also we need $p_q$ memory items to store $\pi_1$.

    (b) We set $\tilde{\pi}_q^N = \tilde{\pi}_q^N + a\pi_1$ which requires $2p_q$ a.o.

    (c) We set $a = (1-\alpha)a$.

    At the end of this step we have. TAO $= 2m_1 n_q + 2n_q + p_q(2m_2 s_q + 2s_q) + n_q + 1 + N(2p_q s_q + 2p_q + 1)$, MM $= p_q + 2 + p_q s_q + p_q + p_q$ and UM $= p_q + p_q s_q + p_q + 1 + p_q$

5. Set $\tilde{\pi}_q^N = \frac{\alpha}{1-(1-\alpha)a}\tilde{\pi}_q^N$. This takes $3 + p_q$ a.o.

So at the end we get TAO $= 2m_1 n_q + 2n_q + p_q(2m_2 s_q + 2s_q) + n_q + 1 + N(2p_q s_q + 2p_q + 1) + p_q + 3 \le 3mp_q s_q + 3p_q s_q N$, MM $= p_q + 2 + p_q s_q + p_q + p_q \le 2p_q s_q$ and UM $= p_q$.

**Remark 1.** *Note that we also can store in the memory all the calculated quantities $\langle \varphi_1, \mathbf{V}_i^q \rangle$ for all $i \in U_q$, $\langle \varphi_2, \mathbf{E}_{ij}^q \rangle$ for all $i, j = 1, \ldots, p_q$ s.t. $i \to j \in E_q$, $\sum_{j \in U_q} \langle \varphi_1, \mathbf{V}_j^q \rangle$, $\sum_{l:i \to l} \langle \varphi_2, \mathbf{E}_{il}^q \rangle$ for the case if we need them later. This requires not more than $n_q + p_q s_q + 1 + p_q$ memory.*

**Lemma A.2.** *Let $q$ in $Q$. Assume that $\pi_1, \pi_2 \in S_{p_q}(1) = \{\pi \in \mathbb{R}_+^{p_q} : \sum_{i=1}^{p_q}[\pi]_i = 1\}$. Assume also that inequality $\|\pi_1 - \pi_2\|_\gamma \le \Delta_1$ holds for some $\gamma \in \{1, 2, \infty\}$. Then*

$$\big|\|(A_q\pi_1)_+\|_2 - \|(A_q\pi_2)_+\|_2\big| \le 2\Delta_1 \sqrt{r_q}, \tag{A.3}$$

$$\|(A_q\pi_1)_+ - (A_q\pi_2)_+\|_\infty \le 2\Delta_1, \tag{A.4}$$

$$\|(A_q\pi_1)_+\|_2 \le \sqrt{r_q}, \tag{A.5}$$

$$\|(A_q\pi_1)_+\|_\infty \le 1. \tag{A.6}$$

**Proof.** Note that for any $\gamma \in \{1, 2, \infty\}$ from the inequality $\|\pi_1 - \pi_2\|_\gamma \le \Delta_1$ it follows that $|[\pi_1]_i - [\pi_2]_i| \le \Delta_1$ for all $i \in 1, \ldots, p_q$. Using Lipschitz continuity with constant 1 of the 2-norm we get

$$\big|\|(A_q\pi_1)_+\|_2 - \|(A_q\pi_2)_+\|_2\big| \le \|(A_q\pi_1)_+ - (A_q\pi_2)_+\|_2 \tag{A.7}$$

Let us fix arbitrary $i \in 1, \ldots, r_q$. By definition the $i$-th row of the matrix $A_q$ contains one 1 and one -1 and all other elements in the row are equal to zero. Let $k : [A_q]_{ik} = 1$, $j : [A_q]_{ij} = -1$. Using Lipschitz continuity with constant 1 of the function $(\cdot)_+$ we obtain

$$\big|[(A_q\pi_1)_+]_i - [(A_q\pi_2)_+]_i\big| \le |[\pi_1]_k - [\pi_1]_j - [\pi_2]_k + [\pi_2]_j| \le 2\Delta_1.$$

Since $i \in 1, \ldots, r_q$ was chosen arbitrary this inequality holds for all $i \in 1, \ldots, r_q$ and using (A.7) we obtain (A.3). Similarly one obtains (A.4).

Now let us fix some $i \in 1, \ldots, r_q$ and again let $k : [A_q]_{ik} = 1$, $j : [A_q]_{ij} = -1$. Then $|[(A_q\pi_1)_+]_i| = |([\pi_1]_k - [\pi_1]_j)_+|$. Since $\pi_1 \in S_{p_q}(1)$ it holds that $[\pi_1]_k - [\pi_1]_j \in [-1, 1]$. Hence $|([\pi_1]_k - [\pi_1]_j)_+| \le 1$. Now (A.5) and (A.6) become obvious.

**Lemma A.3.** *Assume that vectors $\tilde{\pi}_q \in S_{p_q}(1), q \in Q$ satisfy the following inequalities*

$$\|\tilde{\pi}_q - \pi_q(\varphi)\|_\gamma \le \Delta_1, \quad \forall q \in Q,$$

*for some $\gamma \in \{1, 2, \infty\}$. Then*

$$\tilde{f}(\varphi) = \frac{1}{|Q|}\sum_{q=1}^{|Q|} \|(A_q\tilde{\pi}_q)_+\|_2^2 \tag{A.8}$$

*satisfies $|\tilde{f}(\varphi) - f(\varphi)| \le 4r\Delta_1$, where $f(\varphi)$ is defined in (3.1).*

**Proof.** For fixed $q \in Q$ we have

$$\big|\|(A_q\tilde{\pi}_q)_+\|_2^2 - \|(A_q\pi_q(\varphi))_+\|_2^2\big| =$$

$$= \big|\|(A_q\tilde{\pi}_q)_+\|_2 - \|(A_q\pi_q(\varphi))_+\|_2\big| \cdot \big(\|(A_q\tilde{\pi}_q)_+\|_2 + \|(A_q\pi_q(\varphi))_+\|_2\big) \overset{\text{(A.3),(A.5)}}{\le}$$

$$\le 4\Delta_1 r_q \le 4\Delta_1 r.$$

Using (3.1) and (A.8) we obtain the statement of the lemma.

**The proof ot Lemma 1.** Inequality 4.3 follows from Lemma A.1 and Lemma A.3 with $\Delta_1 = \frac{\delta_1}{4r}$ and $\tilde{\pi}_q = \tilde{\pi}_q(\varphi, N)$ for all $q \in Q$.

Let us now estimate the the number of arithmetic operations and memory amount used by the method for calculation of $f(\varphi, \delta_1)$ (4.2).

We use the same notations TAO, MM, UM as in the proof of Lemma A.1.

1. We reserve variable $a$ to store current (in terms of steps in $q$) sum of summands in (4.2), variable $b$ to store next summand in this sum and vector $\pi$ to store the approximation for $\tilde{\pi}_q(\varphi, N)$ for current $q \in Q$. So TAO = 0, MM = UM = $2 + p_q$.

2. For every $q \in Q$ repeat.

   2.1. Set $\pi = \tilde{\pi}_q(\varphi, N)$. According to Lemma A.1 we obtain TAO = $3mp_qs_q + 3p_qs_qN$, MM = $2p_qs_q + p_q + 2$, UM = $p_q + 2$.

   2.2. Calculate $u = (A_q\tilde{\pi}_q(\varphi, N))_+$. This requires additionally $2r_q$ a.o. and $r_q$ memory items.

   2.3. Set $b = \|u\|_2^2$. This requires additionally $2r_q$ a.o.

   2.4. Set $a = a + b$. This requires additionally 1 a.o.

3. Set $a = \frac{1}{|Q|}a$. This requires additionally 1 a.o.

4. At the end we have TAO = $\sum_{q \in Q}(3mp_qs_q + 3p_qs_qN + 4r_q + 1) + 1 \le |Q|(3mps + 3psN + 6r)$, MM = $\max_{q \in Q}(2p_qs_q + p_q) + 2 \le 3ps$, UM = 1.

## A.2 The proof of Lemma 2

We use the following norms on the space of matrices $A \in \mathbb{R}^{n_1 \times n_2}$

$$\|A\|_1 = \max\{\|Ax\|_1 : x \in \mathbb{R}^{n_2}, \|x\|_1 = 1\} = \max_{j=1,\dots,n_2} \sum_{i=1}^{n_1} |a_{ij}|, \tag{A.9}$$

where the 1-norm of the vector $x \in \mathbb{R}^{n_2}$ is $\|x\|_1 = \sum_{i=1}^{n_2} |x_i|$.

$$\|A\|_\infty = \max\{\|Ax\|_\infty : x \in \mathbb{R}^{n_2}, \|x\|_\infty = 1\} = \max_{i=1,\dots,n_1} \sum_{j=1}^{n_2} |a_{ij}|,$$

where the $\infty$-norm of the vector $x \in \mathbb{R}^{n_2}$ is $\|x\|_\infty = \max_{i=1,\dots,n_2} |x_i|$. Note that both matrix norms possess submultiplicative property

$$\|AB\|_1 \le \|A\|_1\|B\|_1, \quad \|AB\|_1 \le \|A\|_\infty\|B\|_\infty \tag{A.10}$$

for any pair of compatible matrices $A, B$.

Let us denote

$$\Pi_q^0(\varphi) = \alpha\frac{d\pi_q^0(\varphi)}{d\varphi^T} + (1-\alpha)\sum_{i=1}^{p_q} \frac{dp_i(\varphi)}{d\varphi^T}[\pi_q(\varphi)]_i. \tag{A.11}$$

**Lemma A.4.** *Let us fix some $q \in Q$. Let $\Pi_q^0(\varphi)$ be defined in (A.11), $\pi_q^0(\varphi)$ be defined in (2.1), $p_i(\varphi)^T$, $i \in 1, \dots, p_q$ be the $i$-th row of the matrix $P_q(\varphi)$ defined in (2.2). Let us denote*

$$\mathbb{V}^q = \sum_{l \in U_q} \mathbf{V}_l^q, \quad \mathbb{E}_i^q = \sum_{l \in N_q(i)} \mathbf{E}_{il}^q,$$

*where $N_q(i) = \{j \in V_q : i \to j \in E_q\}$.*

*Then for the chosen restart probabilities (2.1), transition probabilities (2.2) and set $\Phi = \{\varphi \in \mathbb{R}^m : \|\varphi - \hat{\varphi}\|_2 \le R\}$ in (3.2) the following inequality holds.*

$$\|\Pi_q^0(\varphi)\|_1 \le \alpha\left\|\frac{d\pi_q^0(\varphi)}{d\varphi^T}\right\|_1 + (1-\alpha)\sum_{i=1}^{p_q}\left\|\frac{dp_i(\varphi)}{d\varphi^T}\right\|_1 \le \beta_1 \quad \forall\varphi \in \Phi, \tag{A.12}$$

*where*

$$\beta_1 = 2\alpha\frac{\langle\hat{\varphi}_1, \mathbb{V}^q\rangle + R\|\mathbb{V}^q\|_2}{\left(\langle\hat{\varphi}_1, \mathbb{V}^q\rangle - R\|\mathbb{V}^q\|_2\right)^2}\max_{j\in 1,\dots,m_1}[\mathbb{V}^q]_j +$$

$$+ 2(1-\alpha)\sum_{i=1}^{p_q}\frac{\langle\hat{\varphi}_2, \mathbb{E}_i^q\rangle + R\|\mathbb{E}_i^q\|_2}{\left(\langle\hat{\varphi}_2, \mathbb{E}_i^q\rangle - R\|\mathbb{E}_i^q\|_2\right)^2}\max_{j\in 1,\dots,m_2}[\mathbb{E}_i^q]_j \tag{A.13}$$

*and, $\hat{\varphi}_1 \in \mathbb{R}^{m_1}$ – first $m_1$ components of the vector $\hat{\varphi}$, $\hat{\varphi}_2 \in \mathbb{R}^{m_2}$ – second $m_2$ components of the vector $\hat{\varphi}$.*

**Proof.** First inequality follows from the definition of $\Pi_q^0(\varphi)$ (A.11), triangle inequality for matrix norm and inequalities $|[\pi_q(\varphi)]_i| \le 1$, $i = 1, \dots, p_q$ which hold since $\pi_q(\varphi) \in S_{P_q}(1)$.

Let us now estimate $\left\|\frac{d\pi_q^0(\varphi)}{d\varphi^T}\right\|_1$. Note that $\varphi = (\varphi_1, \varphi_2)^T$. From (2.1) we know that $\frac{d\pi_q^0(\varphi)}{d\varphi_2^T} = 0$. First we estimate the absolute value of the element in the $i$-th row and $j$-th column of the matrix $\frac{d\pi_q^0(\varphi)}{d\varphi_1^T}$. We use

that $\varphi > 0$ for all $\varphi \in \Phi$ and that for all $i \in U_q$ vectors $\mathbf{V}_i^q$ are non-negative and have at least one positive component.

$$\left| \frac{d\left[\pi_q^0(\varphi)\right]_i}{d[\varphi_1]_j} \right| \overset{(2.1)}{=} \left| \frac{1}{\sum_{l \in U_q} \langle \varphi_1, \mathbf{V}_l^q \rangle} \left[\mathbf{V}_i^q\right]_j - \frac{\langle \varphi_1, \mathbf{V}_i^q \rangle}{\left(\sum_{l \in U_q} \langle \varphi_1, \mathbf{V}_l^q \rangle\right)^2} \left[\sum_{l \in U_q} \mathbf{V}_l^q\right]_j \right| =$$

$$= \frac{1}{(\langle \varphi_1, \mathbb{V}^q \rangle)^2} \left| \langle \varphi_1, \mathbb{V}^q \rangle \left[\mathbf{V}_i^q\right]_j - \langle \varphi_1, \mathbf{V}_i^q \rangle \left[\mathbb{V}^q\right]_j \right| \leq$$

$$\leq \frac{1}{\left(\langle \hat{\varphi}_1, \mathbb{V}^q \rangle - R \|\mathbb{V}^q\|_1\right)^2} \left\{ (\langle \varphi_1, \mathbb{V}^q \rangle) \left[\mathbf{V}_i^q\right]_j + \langle \varphi_1, \mathbf{V}_i^q \rangle \left[\mathbb{V}^q\right]_j \right\} \quad \forall \varphi \in \Phi.$$

Here we used the fact that

$$\min_{\varphi \in \Phi} \langle \varphi_1, \mathbb{V}^q \rangle = \min \left\{ \langle \varphi_1, \mathbb{V}^q \rangle : \|\varphi_1 - \hat{\varphi}_1\|_2^2 + \|\varphi_2 - \hat{\varphi}_2\|_2^2 \leq R^2 \right\} = \langle \hat{\varphi}_1, \mathbb{V}^q \rangle - R \|\mathbb{V}^q\|_2 .$$

Then the 1-norm of the $j$-th column of the matrix $\frac{d\pi_q^0(\varphi)}{d\varphi_1^T}$ satisfies for all $\varphi \in \Phi$

$$\sum_{i \in U_q} \left| \frac{d\left[\pi_q^0(\varphi)\right]_i}{d[\varphi_1]_j} \right| \leq \frac{2}{\left(\langle \hat{\varphi}_1, \mathbb{V}^q \rangle - R \|\mathbb{V}^q\|_2\right)^2} \langle \varphi_1, \mathbb{V}^q \rangle \left[\mathbb{V}^q\right]_j \leq$$

$$\leq 2 \frac{\langle \hat{\varphi}_1, \mathbb{V}^q \rangle + R \|\mathbb{V}^q\|_2}{\left(\langle \hat{\varphi}_1, \mathbb{V}^q \rangle - R \|\mathbb{V}^q\|_2\right)^2} \left[\mathbb{V}^q\right]_j .$$

Here we used the fact that

$$\max_{\varphi \in \Phi} \langle \varphi_1, \mathbb{V}^q \rangle = \max \left\{ \langle \varphi_1, \mathbb{V}^q \rangle : \|\varphi_1 - \hat{\varphi}_1\|_2^2 + \|\varphi_2 - \hat{\varphi}_2\|_2^2 \leq R^2 \right\} = \langle \hat{\varphi}_1, \mathbb{V}^q \rangle + R \|\mathbb{V}^q\|_2 .$$

Now we have for all $\varphi \in \Phi$

$$\left\| \frac{d\pi_q^0(\varphi)}{d\varphi^T} \right\|_1 \overset{(A.9)}{=} \max_{j \in 1, \ldots, m_1} \sum_{i \in U_q} \left| \frac{d\left[\pi_q^0(\varphi)\right]_i}{d[\varphi_1]_j} \right| \leq 2 \frac{\langle \hat{\varphi}_1, \mathbb{V}^q \rangle + R \|\mathbb{V}^q\|_2}{\left(\langle \hat{\varphi}_1, \mathbb{V}^q \rangle - R \|\mathbb{V}^q\|_2\right)^2} \max_{j \in 1, \ldots, m_1} \left[\mathbb{V}^q\right]_j .$$

In the same manner we obtain the following estimate for all $\varphi \in \Phi$

$$\left\| \frac{dp_i(\varphi)}{d\varphi^T} \right\|_1 \leq 2 \frac{\langle \hat{\varphi}_2, \mathbb{E}_i^q \rangle + R \|\mathbb{E}_i^q\|_2}{\left(\langle \hat{\varphi}_2, \mathbb{E}_i^q \rangle - R \|\mathbb{E}_i^q\|_2\right)^2} \max_{j \in 1, \ldots, m_2} \left[\mathbb{E}_i^q\right]_j .$$

Finally we have that for all $\varphi \in \Phi$

$$\|\Pi_q^0(\varphi)\|_1 \overset{(A.12)}{\leq} 2\alpha \frac{\langle \hat{\varphi}_1, \mathbb{V}^q \rangle + R \|\mathbb{V}^q\|_2}{\left(\langle \hat{\varphi}_1, \mathbb{V}^q \rangle - R \|\mathbb{V}^q\|_2\right)^2} \max_{j \in 1, \ldots, m_1} \left[\mathbb{V}^q\right]_j +$$

$$+ 2(1 - \alpha) \sum_{i=1}^{p_q} \frac{\langle \hat{\varphi}_2, \mathbb{E}_i^q \rangle + R \|\mathbb{E}_i^q\|_2}{\left(\langle \hat{\varphi}_2, \mathbb{E}_i^q \rangle - R \|\mathbb{E}_i^q\|_2\right)^2} \max_{j \in 1, \ldots, m_2} \left[\mathbb{E}_i^q\right]_j .$$

This finishes the proof.

Let us fix arbitrary $q \in Q$ and assume that we have some approximation $\tilde{\pi}_q \in S_{p_q}(1)$ to the vector $\pi_q(\varphi)$. We consider generalized Method 4.4, 4.5 parametrized by the approximation $\tilde{\pi}_q$

$$\tilde{\Pi}_0(\tilde{\pi}_q) = \alpha \frac{d\pi_q^0(\varphi)}{d\varphi^T} + (1 - \alpha) \sum_{i=1}^{p_q} \frac{dp_i(\varphi)}{d\varphi^T} [\tilde{\pi}_q]_i \tag{A.14}$$

and

$$\Pi_0(\tilde{\pi}_q) = \tilde{\Pi}_0(\tilde{\pi}_q), \quad \Pi_{k+1}(\tilde{\pi}_q) = P_q^T(\varphi) \Pi_k(\tilde{\pi}_q). \tag{A.15}$$

The output is (for some fixed non-negative integer $N_2$)

$$\tilde{\Pi}_q(\varphi, N_2, \tilde{\pi}_q) = \frac{1}{1 - (1 - \alpha)^{N_2 + 1}} \sum_{k=0}^{N_2} (1 - \alpha)^k \Pi_k(\tilde{\pi}_q). \tag{A.16}$$

**Lemma A.5.** *Let us fix some $q \in Q$. Let $\tilde{\Pi}_0(\tilde{\pi}_q)$ be defined in (A.14), where $\pi_q^0(\varphi)$ is defined in (2.1), $p_i(\varphi)^T$, $i \in 1, \ldots, p_q$ is the $i$-th row of the matrix $P_q(\varphi)$ defined in (2.2), $\tilde{\pi}_q \in S_{p_q}(1)$. Let the sequence $\Pi_k(\tilde{\pi}_q)$, $k \geq 0$ be defined in (A.15). Then for the chosen restart probabilities (2.1), transition probabilities (2.2) and set $\Phi$ for all $k \geq 0$ it holds that*

$$\|\Pi_k(\tilde{\pi}_q)\|_1 \leq \beta_1, \quad \forall \varphi \in \Phi, \tag{A.17}$$

$$\left\| \left[ P_q^T(\varphi) \right]^k \Pi_q^0(\varphi) \right\|_1 \leq \beta_1, \quad \forall \varphi \in \Phi. \tag{A.18}$$

*Here $\Pi_q^0(\varphi)$ is defined in (A.11), $\beta_1$ is defined in (A.13).*

**Proof.** Similarly as it was done in Lemma A.4 one can prove that $\|\tilde{\Pi}_0(\tilde{\pi}_q)\|_1 \leq \beta_1$. Note that all elements of the matrix $P_q^T(\varphi)$ are non-negative for all $\varphi \in \Phi$. Also the matrix $P_q(\varphi)$ is row-stochastic: $P_q(\varphi)e_{p_q} = e_{p_q}$, where $e_{p_q} \in \mathbb{R}^{p_q}$ is the vector of all ones. Hence maximum 1-norm of the column of $P_q^T(\varphi)$ is equal to 1 and $\|P_q^T(\varphi)\|_1 = 1$. Using the submultiplicative property (A.10) of the matrix 1-norm we obtain by induction that

$$\|\Pi_{k+1}(\tilde{\pi}_q)\|_1 = \|P_q^T(\varphi)\Pi_k(\tilde{\pi}_q)\|_1 \overset{\text{(A.10)}}{\leq} \|P_q^T(\varphi)\|_1 \|\Pi_k(\tilde{\pi}_q)\|_1 \leq \beta_1.$$

Inequality A.18 is proved in the same way using the Lemma A.4 as the induction basis.

**Lemma A.6.** *Let the assumptions of Lemma A.5 hold. Then for any $N > 1$*

$$\|\tilde{\Pi}_q(\varphi, N, \tilde{\pi}_q)\|_1 \leq \frac{\beta_1}{\alpha}, \quad \forall \varphi \in \Phi, \tag{A.19}$$

*where $\tilde{\Pi}_q(\varphi, N, \tilde{\pi}_q)$ is defined in (A.16), $\beta_1$ is defined in (A.13).*

**Proof.** Using the triangle inequality for the matrix 1-norm we obtain

$$\|\tilde{\Pi}_q(\varphi, N, \tilde{\pi}_q)\|_1 \overset{\text{(A.16)}}{=} \left\| \frac{1}{1 - (1-\alpha)^{N+1}} \sum_{k=0}^{N} (1-\alpha)^k \Pi_k(\tilde{\pi}_q) \right\|_1 \leq$$

$$\frac{1}{1 - (1-\alpha)^{N+1}} \sum_{k=0}^{N} (1-\alpha)^k \|\Pi_k(\tilde{\pi}_q)\|_1 \overset{\text{(A.17)}}{\leq} \frac{\beta_1}{\alpha}.$$

**Lemma A.7.** *Let us fix some $q \in Q$. Let $\Pi_q^0(\varphi)$ be defined in (A.11) and $\tilde{\Pi}_0(\tilde{\pi}_q)$ be defined in (A.14), where $\pi_q^0(\varphi)$ is defined in (2.1), $p_i(\varphi)^T$, $i \in 1, \ldots, p_q$ is the $i$-th row of the matrix $P_q(\varphi)$ defined in (2.2). Assume that the vector $\tilde{\pi}_q \in S_{p_q}(1)$ satisfies*

$$\|\tilde{\pi}_q - \pi_q(\varphi)\|_1 \leq \Delta_1. \tag{A.20}$$

*Then for the chosen restart prbabilities (2.1), transition probabilities (2.2) and set $\Phi$ it holds that.*

$$\|\tilde{\Pi}_0(\tilde{\pi}_q) - \Pi_q^0(\varphi)\|_1 \leq \beta_1 \Delta_1 \quad \forall \varphi \in \Phi, \tag{A.21}$$

*where $\beta_1$ is defined in (A.13).*

**Proof.**

$$\|\tilde{\Pi}_0(\tilde{\pi}_q) - \Pi_q^0(\varphi)\|_1 \overset{\text{(3.3),(A.14)}}{=} (1-\alpha) \left\| \sum_{i=1}^{p_q} \frac{dp_i(\varphi)}{d\varphi^T} \left( [\tilde{\pi}_q]_i - [\pi_q(\varphi)]_i \right) \right\|_1 \leq$$

$$\leq (1-\alpha) \sum_{i=1}^{p_q} \left\| \frac{dp_i(\varphi)}{d\varphi^T} \right\|_1 |[\tilde{\pi}_q]_i - [\pi_q(\varphi)]_i| \overset{\text{(A.12),(A.20)}}{\leq} \beta_1 \Delta_1.$$

**Lemma A.8.** *Let us fix some $q \in Q$. Let $\tilde{\Pi}_q(\varphi, N, \tilde{\pi}_q)$ be cdefined in (A.16) and $\frac{d\pi_q(\varphi)}{d\varphi^T}$ be given in (3.3), where $\pi_q^0(\varphi)$ is defined in (2.1), $p_i(\varphi)^T$, $i \in 1, \ldots, p_q$ is the $i$-th row of the matrix $P_q(\varphi)$ defined in (2.2). Assume that the vector $\tilde{\pi} \in S_{p_q}(1)$ in (A.14) satisfies $\|\tilde{\pi} - \pi_q(\varphi)\|_1 \leq \Delta_1$. Then for the chosen restart prbabilities (2.1), transition probabilities (2.2) and set $\Phi$, for all $N > 1$ it holds that*

$$\left\| \tilde{\Pi}_q(\varphi, N, \tilde{\pi}_q) - \frac{d\pi_q(\varphi)}{d\varphi^T} \right\|_1 \leq \frac{\beta_1 \Delta_1}{\alpha} + \frac{2\beta_1}{\alpha}(1-\alpha)^{N+1}, \quad \forall \varphi \in \Phi, \tag{A.22}$$

*where $\beta_1$ is defined in (A.13).*

**Proof.** Using (A.21) as the induction basis and making the same arguments as in the proof of the Lemma A.5 we obtain for every $k \geq 0$

$$\left\| \Pi_{k+1}(\tilde{\pi}_q) - [P_q^T(\varphi)]^{k+1} \Pi_q^0(\varphi) \right\|_1 = \left\| P_q^T(\varphi) \left( \Pi_k(\tilde{\pi}_q) - [P_q^T(\varphi)]^k \Pi_q^0(\varphi) \right) \right\|_1 \overset{\text{(A.10)}}{\leq}$$

$$\leq \left\| P_q^T(\varphi) \right\|_1 \left\| \Pi_k(\tilde{\pi}_q) - [P_q^T(\varphi)]^k \Pi_q^0(\varphi) \right\|_1 \leq \beta_1 \Delta_1.$$

Equation 3.3 can be rewritten in the following way

$$\frac{d\pi_q(\varphi)}{d\varphi^T} = \left[I - (1-\alpha)P_q^T(\varphi)\right]^{-1}\Pi_q^0(\varphi) = \sum_{k=0}^{\infty}(1-\alpha)^k\left[P_q^T(\varphi)\right]^k\Pi_q^0(\varphi). \tag{A.23}$$

Using this equality and the previous inequality we obtain

$$\left\|\sum_{k=0}^{\infty}(1-\alpha)^k\Pi_k(\tilde{\pi}_q) - \frac{d\pi_q(\varphi)}{d\varphi^T}\right\|_1 = \left\|\sum_{k=0}^{\infty}(1-\alpha)^k\Pi_k(\tilde{\pi}_q) - \sum_{k=0}^{\infty}(1-\alpha)^k\left[P_q^T(\varphi)\right]^k\Pi_q^0(\varphi)\right\|_1 \le$$

$$\le \sum_{k=0}^{\infty}(1-\alpha)^k\left\|\Pi_k(\tilde{\pi}_q) - \left[P_q^T(\varphi)\right]^k\Pi_q^0(\varphi)\right\|_1 \le \frac{\beta_1\Delta_1}{\alpha}. \tag{A.24}$$

On the other hand

$$\left\|\tilde{\Pi}_q(\varphi, N, \tilde{\pi}_q) - \sum_{k=0}^{\infty}(1-\alpha)^k\Pi_k(\tilde{\pi}_q)\right\|_1 \overset{(A.16)}{=}$$

$$= \left\|\frac{1}{1-(1-\alpha)^{N+1}}\sum_{k=0}^{N}(1-\alpha)^k\Pi_k(\tilde{\pi}_q) - \sum_{k=0}^{\infty}(1-\alpha)^k\Pi_k(\tilde{\pi}_q)\right\|_1 =$$

$$= \left\|\frac{(1-\alpha)^{N+1}}{1-(1-\alpha)^{N+1}}\sum_{k=0}^{N}(1-\alpha)^k\Pi_k(\tilde{\pi}_q) - \sum_{k=N+1}^{\infty}(1-\alpha)^k\Pi_k(\tilde{\pi}_q)\right\|_1 \overset{(A.17)}{\le}$$

$$\le \frac{\beta_1(1-\alpha)^{N+1}}{1-(1-\alpha)^{N+1}}\sum_{k=0}^{N}(1-\alpha)^k + \beta_1\sum_{k=N+1}^{\infty}(1-\alpha)^k = \frac{2\beta_1}{\alpha}(1-\alpha)^{N+1}.$$

This inequality together with (A.24) gives (A.22) by the triangle inequality.

**Lemma A.9.** *Assume that for every $q \in Q$ the approximation $\tilde{\pi}_q(\varphi)$ to the ranking vector, satisfying $\|\tilde{\pi}_q(\varphi) - \pi_q(\varphi)\|_1 \le \Delta_1$, is available. Assume that for every $q \in Q$ the approximation $\tilde{\Pi}_q(\varphi)$ to the full derivative of ranking vector $\frac{d\pi_q(\varphi)}{d\varphi^T}$ as solution of (3.3), satisfying*

$$\left\|\tilde{\Pi}_q(\varphi) - \frac{d\pi_q(\varphi)}{d\varphi^T}\right\|_1 \le \Delta_2$$

*is available. Let us define*

$$\tilde{\nabla}f(\varphi) = \frac{2}{|Q|}\sum_{q=1}^{|Q|}\left(\tilde{\Pi}_q(\varphi)\right)^T A_q^T(A_q\tilde{\pi}_q(\varphi))_+. \tag{A.25}$$

*Then*

$$\left\|\tilde{\nabla}f(\varphi) - \nabla f(\varphi)\right\|_{\infty} \le 2r\Delta_2 + 4r\Delta_1\max_{q\in Q}\left\|\tilde{\Pi}_q(\varphi)\right\|_1, \tag{A.26}$$

*where $\nabla f(\varphi)$ is the gradient (3.4) of the function $f(\varphi)$ (3.1).*

**Proof.** Let us fix any $q \in Q$. Then we have

$$\left\|\left(\tilde{\Pi}_q(\varphi)\right)^T A_q^T(A_q\tilde{\pi}_q(\varphi))_+ - \left(\frac{d\pi_q(\varphi)}{d\varphi^T}\right)^T A_q^T(A_q\pi_q(\varphi))_+\right\|_{\infty} \le$$

$$\le \left\|\left(\tilde{\Pi}_q(\varphi)\right)^T A_q^T(A_q\tilde{\pi}_q(\varphi))_+ - \left(\tilde{\Pi}_q(\varphi)\right)^T A_q^T(A_q\pi_q(\varphi))_+\right\|_{\infty} +$$

$$+ \left\|\left(\tilde{\Pi}_q(\varphi)\right)^T A_q^T(A_q\pi_q(\varphi))_+ - \left(\frac{d\pi_q(\varphi)}{d\varphi^T}\right)^T A_q^T(A_q\pi_q(\varphi))_+\right\|_{\infty} \overset{(A.10)}{\le}$$

$$\le \left\|\tilde{\Pi}_q(\varphi)\right\|_1\|A_q\|_1\|(A_q\pi_q(\varphi))_+ - (A_q\tilde{\pi}_q(\varphi))_+\|_{\infty} +$$

$$+ \left\|\tilde{\Pi}_q(\varphi) - \frac{d\pi_q(\varphi)}{d\varphi^T}\right\|_1\|A_q\|_1\|(A_q\pi_q(\varphi))_+\|_{\infty} \overset{(A.4),(A.6)}{\le} \left\|\tilde{\Pi}_q(\varphi)\right\|_1 \cdot r \cdot 2\Delta_1 + \Delta_2 \cdot r \cdot 1.$$

Here we used that $A_q \in \mathbb{R}^{r_q \times p_q}$ and its elements are either 0 or 1 and the fact that $r_q \le r$ for all $q \in Q$, and that for any matrix $M \in \mathbb{R}^{n_1 \times n_2}$ $\|M^T\|_{\infty} = \|M\|_1$.

Using this inequality and definitions (3.4), (A.25) we obtain (A.26).

**Proof of Lemma 2**

Let us first prove Inequality 4.8. According to Lemma A.1 calculated vector $\tilde{\pi}_q(\varphi, N_1)$ satisfies

$$\|\tilde{\pi}_q(\varphi, N_1) - \pi_q(\varphi)\|_1 \leq \frac{\alpha \delta_2}{12\beta_1 r}, \quad \forall q \in Q. \tag{A.27}$$

This together with Lemma A.8 with $\tilde{\pi}_q(\varphi, N_1)$ in the role of $\tilde{\pi}_q$ for all $q \in Q$ gives since $\tilde{\Pi}_q(\varphi, N_2) = \tilde{\Pi}_q(\varphi, N_2, \tilde{\pi}_q(\varphi, N_1))$

$$\left\| \tilde{\Pi}_q(\varphi, N_2) - \frac{d\pi_q(\varphi)}{d\varphi^T} \right\|_1 \leq \frac{\beta_1 \frac{\alpha \delta_2}{12\beta_1 r}}{\alpha} + \frac{2\beta_1}{\alpha}(1-\alpha)^{N_2+1} \leq \frac{\delta_2}{12r} + \frac{\beta_1}{\alpha}\frac{\alpha \delta_2}{4\beta_1 r} = \frac{\delta_2}{3r}$$

This inequality together with (A.27), Lemma A.6 with $\tilde{\pi}_q(\varphi, N_1)$ in the role of $\tilde{\pi}_q$ for all $q \in Q$ and Lemma A.9 with $\tilde{\pi}_q(\varphi, N_1)$ in the role of $\tilde{\pi}_q(\varphi)$ and $\tilde{\Pi}_q(\varphi, N_2)$ in the role of $\tilde{\Pi}_q(\varphi)$ for all $q \in Q$ gives

$$\|\tilde{g}(\varphi, \delta_2) - \nabla f(\varphi)\|_\infty \leq 2r\frac{\delta_2}{3r} + 4r\frac{\alpha \delta_2}{12\beta_1 r}\frac{\beta_1}{\alpha} = \delta_2.$$

Let us now estimate number of a.o. and memory which is needed to calculate $\tilde{g}(\varphi, \delta_2)$. We use the same notations TAO, MM, UM as in the proof of Lemma A.1.

1. We reserve vector $g_1 \in \mathbb{R}^m$ to store current (in terms of steps in $q$) approximation of $\tilde{g}(\varphi, \delta_2)$ and $g_2 \in \mathbb{R}^m$ to store next summand in the sum (4.7). So TAO $= 0$, MM $=$ UM $= 2m$.

2. For every $q \in Q$ repeat.

   2.1. Set $\pi = \tilde{\pi}_q(\varphi, N_1)$. Also save in memory $\langle \varphi_1, \mathbf{V}_j^q \rangle$ for all $j \in U_q$ ; $\langle \varphi_2, \mathbf{E}_{il}^q \rangle$ for all $i \in V_q$, $l : i \to l$; $\sum_{j \in U_q} \langle \varphi_1, \mathbf{V}_j^q \rangle$ and $\sum_{l:i \to l} \langle \varphi_2, \mathbf{E}_{il}^q \rangle$ for all $i \in V_q$ and the matrix $P_q(\varphi)$. All this data was calculated during the calculation of $\tilde{\pi}_q(\varphi, N_1)$, see the proof of Lemma A.1. According to Lemma A.1 and memory used to save the listed objects we obtain TAO $= 3mp_q s_q + 3p_q s_q N_1$, MM $= 2m + 2p_q s_q + n_q + p_q s_q + 1 + p_q \leq 2m + 4p_q s_q$, UM $= 2m + p_q + n_q + p_q s_q + 1 + p_q + p_q s_q \leq 2m + 3p_q s_q$.

   2.2. Now we need to calculate $\tilde{\Pi}_q(\varphi, N_2)$. We reserve variables $G_t, G_1, G_2 \in \mathbb{R}^{p_q \times m}$ to store respectively sum in (4.5) , $\Pi_k$, $\Pi_{k+1}$ for current $k \in 1, \ldots, N_2$. Hence TAO $= 3mp_q s_q + 3p_q s_q N_1$, MM $= 2m + 4p_q s_q + 3mp_q$, UM $= 2m + 3p_q s_q + 3mp_q$.

   2.2.1. First iteration of this method requires to calculate $\tilde{\Pi}_0 = \alpha \frac{d\pi_q^0(\varphi)}{d\varphi^T} + (1 - \alpha) \sum_{i=1}^{p_q} \frac{dp_i(\varphi)}{d\varphi^T} [\tilde{\pi}_q(\varphi, N_1)]_i$.

   2.2.1.1. We first calculate $G_1 = \alpha \frac{d\pi_q^0(\varphi)}{d\varphi^T}$. In accordance to its definition (2.1) one has for all $i \in U_q, l = 1, \ldots, m_1$

   $$\left[ \frac{\alpha[\pi_q^0]_i}{d\varphi} \right]_l = \left[ \frac{\alpha \mathbf{V}_i^q}{\sum_{j \in U_q} \langle \varphi_1, \mathbf{V}_j^q \rangle} - \frac{\alpha \langle \varphi_1, \mathbf{V}_i^q \rangle}{\left( \sum_{j \in U_q} \langle \varphi_1, \mathbf{V}_j^q \rangle \right)^2} \sum_{j \in U_q} \mathbf{V}_j^q \right]_l$$

   and $\left[ \frac{\alpha[\pi_q^0]_i}{d\varphi} \right]_l = 0$ for $l = m_1 + 1, \ldots, m$. We set $a = \frac{\alpha}{\sum_{j \in U_q} \langle \varphi_1, \mathbf{V}_j^q \rangle}$ and $b = \frac{a}{\sum_{j \in U_q} \langle \varphi_1, \mathbf{V}_j^q \rangle}$, $v = \sum_{j \in U_q} \mathbf{V}_j^q$. This requires $2 + m_1 n_q$ a.o. and $2 + m_1$ memory items. Now the calculation of all non-zero elements of $\alpha \frac{d\pi_q^0(\varphi)}{d\varphi^T}$ takes $4m_1 n_q$ a.o. since for fixed $i, l$ we need 4 a.o. We obtain TAO $= 3mp_q s_q + 3p_q s_q N_1 + 5m_1 n_q + 2$, MM $= 2m + 4p_q s_q + 3mp_q + m_1 + 2$, UM $= 2m + 3p_q s_q + 3mp_q$.

   2.2.1.2. Now we calculate $\tilde{\Pi}_0$. For every $i = 1, \ldots, p_q$ the matrix $(1 - \alpha)\frac{dp_i(\varphi)}{d\varphi^T}[\tilde{\pi}_q(\varphi, N_1)]_i \in \mathbb{R}^{p_q \times m}$ is calculated in the same way as the matrix $\alpha \frac{d\pi_q^0(\varphi)}{d\varphi^T}$ with obvious modifications due to $\frac{dp_i(\varphi)}{d\varphi_1^T} = 0$ and number of non-zero elements in vector $p_i(\varphi)$ is not more than $s_q$. We also use additional a.o. number and memory amount to calculate and save $(1 - \alpha)[\tilde{\pi}_q(\varphi, N_1)]_i$. We save the result for current $i$ in $G_2$. So for fixed $i$ we need additionally $3 + 5m_2 s_q$ a.o and $3 + m_2$ memory items. Also on every step we set $G_1 = G_1 + G_2$ which requires not more than $m_2 s_q$ a.o. since at every step $G_2$ has not more than $m_2 s_q$ non-zero elements. We set $G_t = G_1$. Note that $G_t$ always has a block of $(p_q - n_q) \times m_1$ zero elements and hence has not more than $m_2 p_q + m_1 n_q$ non-zero

elements. At the end we obtain TAO $= 3mp_qs_q + 3p_qs_qN_1 + 5m_1n_q + 2 + p_q(3 + 5m_2s_q + m_2s_q) + m_2p_q + m_1n_q$, MM $= 2m + 4p_qs_q + 3mp_q + m_1 + 2 + m_2 + 3 \leq 3m + 4p_qs_q + 3mp_q + 5$, UM $= 2m + p_qs_q + 3mp_q + p_q$ (since we need to store in memory only $g_1, g_2, G_t, G_1, G_2, P_q^T(\varphi), \pi$).

2.2.2. Set $a = (1 - \alpha)$.

2.2.3. For every step $k$ from 1 to $N_2$

    2.2.3.1. We set $G_2 = P_q^T(\varphi)G_1$. In this pperation potentially each of $p_qs_q$ elements of matrix $P_q^T(\varphi)$ needs to be multiplied my $m$ elements of matrix $G_1$ and this multiplication is coupled with one addition. So in total we need $2mp_qs_q$ a.o.

    2.2.3.2. We set $G_t = G_t + aG_1$. This requires $2m_1n_q + 2m_2p_q$ a.o.

    2.2.3.3. We set $a = (1 - \alpha)a$.

    2.2.3.4. In total every step requires not more than $2mp_qs_q + 2m_1n_q + 2m_2p_q + 1$ a.o.

2.2.4. At the end o this stage we have. TAO $= 3mp_qs_q + 3p_qs_qN_1 + 5m_1n_q + 2 + p_q(3 + 5m_2s_q + m_2s_q) + m_2p_q + m_1n_q + N_2(2mp_qs_q + 2m_1n_q + 2m_2p_q + 1)$, MM $= 3m + 4p_qs_q + 3mp_q + 5$, UM $= 2m + mp_q + p_q$ (since we need to store in memory only $g_1, g_2, G_t, \pi$).

2.2.5. Set $G_t = \frac{\alpha}{1-(1-\alpha)a}G_t$. This takes $3 + m_2p_q + m_1n_q$ a.o.

2.2.6. At the end o this stage we have. TAO $= 3mp_qs_q + 3p_qs_qN_1 + 5m_1n_q + 2 + p_q(3 + 5m_2s_q + m_2s_q) + m_2p_q + m_1n_q + N_2(2mp_qs_q + 2m_1n_q + 2m_2p_q + 1) + 3 + m_2p_q + m_1n_q$, MM $= 3m + 4p_qs_q + 3mp_q + 5$, UM $= 2m + mp_q + p_q$ (since we need to store in memory only $g_1, g_2, G_t, \pi$).

2.3. Calculate $u = (A_q\tilde{\pi}_q(\varphi, N_1))_+$. This requires additionally $2r_q$ a.o. and $r_q$ memory.

2.4. Calculate $\pi = A_q^Tu$. This requires additionally $4r_q$ a.o.

2.5. Calculate $g_2 = G_t^T\pi$. This requires additionally $2m_1n_q + 2m_2p_q$ a.o.

2.6. Set $g_1 = g_1 + g_2$. This requires additionally $m$ a.o.

2.7. At the end we have TAO $= 3mp_qs_q + 3p_qs_qN_1 + 5m_1n_q + 2 + p_q(3 + 5m_2s_q + m_2s_q) + m_2p_q + m_1n_q + N_2(2mp_qs_q + 2m_1n_q + 2m_2p_q + 1) + 3 + m_2p_q + m_1n_q + 7r_q + 2m_1n_q + 2m_2p_q + m$, MM $= 3m + 4p_qs_q + 3mp_q + 5 + r_q$, UM $= 2m$ (since we need to store in memory only $g_1, g_2$).

3. Set $g_1 = \frac{2}{|Q|}g_1$. This requires additionally $m + 1$ a.o.

4. At the end we have TAO $= \sum_{q \in Q}(3mp_qs_q + 3p_qs_qN_1 + 5m_1n_q + 2 + p_q(3 + 5m_2s_q + m_2s_q) + m_2p_q + m_1n_q + N_2(2mp_qs_q + 2m_1n_q + 2m_2p_q + 1) + 3 + m_2p_q + m_1n_q + 6r_q + 2m_1n_q + 2m_2p_q + m) + m + 1 \leq |Q|(10mps + 3psN_1 + 3mpsN_2 + 7r)$, MM $= 3m + 5 + \max_{q \in Q}(4p_qs_q + 3mp_q + r_q) \leq 4ps + 4mp + r$, UM $= m$ (since we need to store in memory only $g_1$).

# B   Missed proofs for Section 5

Consider smoothed counterpart of the function $f(x)$:

$$f_\tau(x) = \mathbb{E}f(x + \tau\zeta) = \frac{1}{V_{\mathcal{B}}}\int_{\mathcal{B}} f(x + \tau\zeta)d\zeta,$$

where $\zeta$ is uniformly distributed over unit ball $\mathcal{B} = \{t \in \mathbb{R}^m : \|t\|_2 \leq 1\}$ random vector, $V_{\mathcal{B}}$ is the volume of the unit ball $\mathcal{B}$, $\tau \geq 0$ is a smoothing parameter. This type of smoothing is well known.

It is easy to show that

- If $f$ is convex, then $f_\tau$ is also convex
- If $f \in C_L^{1,1}(\|\cdot\|_2)$, then $f_\tau \in C_L^{1,1}(\|\cdot\|_2)$.
- If $f \in C_L^{1,1}(\|\cdot\|_2)$, then $f(x) \leq f_\tau(x) \leq f(x) + \frac{L\tau^2}{2}$ for all $x \in \mathbb{R}^m$.

The random gradient-free oracle is usually defined as follows

$$g_\tau(x) = \frac{m}{\tau}(f(x + \tau\xi) - f(x))\xi,$$

where $\xi$ is uniformly distributed vector over the unit sphere $\mathcal{S} = \{t \in \mathbb{R}^m : \|t\|_2 = 1\}$. It can be shown that $\mathbb{E}g_\tau(x) = \nabla f_\tau(x)$. Since we can use only inexact zeroth-order oracle we also define the counterpart of the above random gradient-free oracle which can be really computed:

$$g_\tau(x, \delta) = \frac{m}{\tau}(\tilde{f}(x + \tau\xi, \delta) - \tilde{f}(x, \delta))\xi.$$

The idea is to use gradient-type method with oracle $g_\tau(x, \delta)$ instead of the real gradient in order to minimize $f_\tau(x)$. Since $f_\tau(x)$ is uniformly close to $f(x)$ we can obtain a good approximation to the minimum value of $f(x)$.

We will need the following lemma.

**Lemma B.10.** *Let $\xi$ be random vector uniformly distributed over the unit sphere $S \in \mathbb{R}^m$. Then*

$$\mathbb{E}_\xi(\langle \nabla f(x), \xi \rangle)^2 = \frac{1}{m} \|\nabla f(x)\|_2^2. \tag{B.1}$$

**Proof.** We have $\mathbb{E}_\xi(\langle \nabla f(x), \xi \rangle)^2 = \frac{1}{S_m(1)} \int_S (\langle \nabla f(x), \xi \rangle)^2 d\sigma(\xi)$, where $S_m(r)$ is the volume of the unit sphere which is the border of the ball in $\mathbb{R}^m$ with radius $r$, $\sigma(\xi)$ is unnormalized spherical measure. Note that $S_m(r) = S_m(1)r^{m-1}$. Let $\varphi$ be the angle between $\nabla f(x)$ and $\xi$. Then

$$\frac{1}{S_m(1)} \int_S (\langle \nabla f(x), \xi \rangle)^2 d\sigma(\xi) = \frac{1}{S_m(1)} \int_0^\pi \|\nabla f(x)\|_2^2 \cos^2 \varphi S_{m-1}(\sin \varphi) d\varphi =$$

$$= \frac{S_{m-1}(1)}{S_m(1)} \|\nabla f(x)\|_2^2 \int_0^\pi \cos^2 \varphi \sin^{m-2} \varphi d\varphi$$

First changing the variable using equation $x = \cos \varphi$, and then $t = x^2$, we obtain

$$\int_0^\pi \cos^2 \varphi \sin^{m-2} \varphi d\varphi = \int_{-1}^1 x^2(1-x^2)^{(m-3)/2} dx = \int_0^1 t^{1/2}(1-t)^{(m-3)/2} dt =$$

$$= B\left(\frac{3}{2}, \frac{m-1}{2}\right) = \frac{\sqrt{\pi}\Gamma\left(\frac{m-1}{2}\right)}{2\Gamma\left(\frac{m+2}{2}\right)},$$

where $\Gamma(\cdot)$ is the Gamma-function and $B$ is the Beta-function. Also we have

$$\frac{S_{m-1}(1)}{S_m(1)} = \frac{m-1}{m\sqrt{\pi}} \frac{\Gamma\left(\frac{m+2}{2}\right)}{\Gamma\left(\frac{m+1}{2}\right)}.$$

Finally using the relation $\Gamma(m+1) = m\Gamma(m)$, we obtain

$$\mathbb{E}(\langle \nabla f(x), \xi \rangle)^2 = \|\nabla f(x)\|_2^2 \left(1 - \frac{1}{m}\right) \frac{\Gamma\left(\frac{m-1}{2}\right)}{2\Gamma\left(\frac{m+1}{2}\right)} = \|\nabla f(x)\|_2^2 \left(1 - \frac{1}{m}\right) \frac{\Gamma\left(\frac{m-1}{2}\right)}{2\frac{m-1}{2}\Gamma\left(\frac{m-1}{2}\right)} =$$

$$= \frac{1}{m} \|\nabla f(x)\|_2^2$$

**Lemma B.11.** *Let $f \in C_L^{1,1}(\|\cdot\|_2)$. Then, for any $x, y \in \mathbb{R}^m$,*

$$\mathbb{E}\|g_\tau(x,\delta)\|_2^2 \leq m^2\tau^2 L^2 + 4m\|\nabla f(x)\|_2^2 + \frac{8\delta^2 m^2}{\tau^2} \tag{B.2}$$

$$-\mathbb{E}\langle g_\tau(x,\delta), x - y \rangle \leq -\langle \nabla f_\tau(x), x - y \rangle + \frac{\delta m}{\tau}\|x - y\|_2. \tag{B.3}$$

**Proof.** Using (5.1) we obtain

$$(\tilde{f}(x + \tau\xi, \delta) - \tilde{f}(x, \delta))^2 =$$

$$(f(x + \tau\xi) - f(x) - \tau\langle \nabla f(x), \xi \rangle + \tau\langle \nabla f(x), \xi \rangle + \tilde{\delta}(x + \tau\xi) - \tilde{\delta}(x))^2 \leq$$

$$2(f(x + \tau\xi) - f(x) - \tau\langle \nabla f(x), \xi \rangle + \tau\langle \nabla f(x), \xi \rangle)^2 + 2(\tilde{\delta}(x + \tau\xi) - \tilde{\delta}(x))^2 \leq$$

$$4\left(\frac{\tau^2}{2}L\|\xi\|^2\right)^2 + 4\tau^2(\langle \nabla f(x), \xi \rangle)^2 + 8\delta^2 = \tau^4 L^2\|\xi\|^4 + 4\tau^2(\langle \nabla f(x), \xi \rangle)^2 + 8\delta^2$$

Using (B.1), we get

$$\mathbb{E}_\xi\|g_\tau(x,\delta)\|_2^2 \leq \frac{m^2}{\tau^2 V_s} \int_S \left(\tau^4 L^2\|\xi\|^4 + 4\tau^2(\langle \nabla f(x), \xi \rangle)^2 + 8\delta^2\right)\|\xi\|_2^2 d\sigma(\xi) =$$

$$= m^2\tau^2 L^2 + 4m\|\nabla f(x)\|_2^2 + \frac{8\delta^2 m^2}{\tau^2}.$$

Using the equality $\mathbb{E}_\xi g_\tau(x) = \nabla f_\tau(x)$, we have

$$-\mathbb{E}_\xi\langle g_\tau(x,\delta), x - y \rangle = -\frac{m}{\tau V_s} \int_S (f_\delta(x + \tau\xi) - f_\delta(x))\langle \xi, x - y \rangle d\sigma(\xi) =$$

$$= -\frac{m}{\tau V_s} \int_S (f(x + \tau\xi) - f(x))\langle \xi, x - y \rangle d\sigma(\xi) -$$

$$-\frac{m}{\tau V_s} \int_S (\tilde{\delta}(x + \tau\xi) - \tilde{\delta}(x))\langle \xi, x - y \rangle d\sigma(\xi) \leq -\langle \nabla f_\tau(x), x - y \rangle + \frac{\delta m}{\tau}\|x - y\|.$$

Let us denote $\psi_0 = f(x_0)$, and $\psi_k = \mathbb{E}_{\Xi_{k-1}} f(x_k)$, $k \geq 1$.

We say that the smooth function is $\mu$-strongly convex (or strongly convex with parameter $\mu \geq 0$) if and only if for any $x, y \in \mathbb{R}^m$ it holds that

$$f(x) \geq f(y) + \langle \nabla f(y), x - y \rangle + \frac{\mu}{2} \|x - y\|^2. \tag{B.4}$$

**Theorem 1** (extended) *Let $f \in C_L^{1,1}(\| \cdot \|_2)$ and convex. Assume that $x^* \in \mathrm{int} X$ and the sequence $x_k$ be generated by Algorithm 1 with $h = \frac{1}{8mL}$. Then for any $M \geq 0$, we have*

$$\mathbb{E}_{\mathcal{U}_{M-1}} f(\hat{x}_M) - f^* \leq \frac{8mLD^2}{M+1} + \frac{\tau^2 L(m+8)}{8} + \frac{\delta mD}{4\tau} + \frac{\delta^2 m}{L\tau^2},$$

*where $f^*$ is the solution of the problem $\min_{x \in X} f(x)$. If, moreover, $f$ strongly convex with constant $\mu$, then*

$$\psi_M - f^* \leq \frac{1}{2} L \left( \delta_\tau + \left(1 - \frac{\mu}{16mL}\right)^M (D^2 - \delta_\tau) \right), \tag{B.5}$$

*where $\delta_\tau = \frac{\tau^2 L(m+8)}{4\mu} + \frac{4m\delta D}{\mu\tau} + \frac{2m\delta^2}{\mu\tau^2 L}$.*

**Proof.** We extend the proof in [16] for the case of constrained optimization, randomization on a sphere (instead of randomization based on normal distribution) and for the case when one can calculate the function value only with some error of unknown nature.

Consider the point $x_k$, $k \geq 0$ generated by the method on the $k$-th iteration. Denote $r_k = \|x_k - x^*\|_2$. Note that $r_k \leq D$. We have:

$$r_{k+1}^2 = \|x_{k+1} - x^*\|_2^2 \leq \|x_k - x^* - hg_\tau(x_k, \delta)\|_2^2 =$$
$$= \|x_k - x^*\|_2^2 - 2h\langle g_\tau(x_k, \delta), x_k - x^* \rangle + h^2 \|g_\tau(x_k, \delta)\|_2^2.$$

Taking the expectation with respect to $\xi_k$ we get

$$\mathbb{E}_{\xi_k} r_{k+1}^2 \overset{(B.2),(B.3)}{\leq} r_k^2 - 2h\langle \nabla f_\tau(x_k), x_k - x^* \rangle + \frac{2\delta mh}{\tau} r_k +$$

$$+ h^2 \left( m^2\tau^2 L^2 + 4m\|\nabla f(x_k)\|_2^2 + \frac{8\delta^2 m^2}{\tau^2} \right) \leq$$

$$\leq r_k^2 - 2h(f(x_k) - f_\tau(x^*)) + \frac{\delta mhD}{4\tau} +$$

$$+ h^2 \left( m^2\tau^2 L^2 + 8mL(f(x_k) - f^*) + \frac{8\delta^2 m^2}{\tau^2} \right) \leq$$

$$\leq r_k^2 - 2h(1 - 4hmL)(f(x_k) - f^*) + \frac{\delta mhD}{4\tau} +$$

$$+ m^2 h^2 \tau^2 L^2 + hL\tau^2 + \frac{8\delta^2 m^2 h^2}{\tau^2} \leq$$

$$\leq r_k^2 + \frac{D\delta}{4\tau L} - \frac{f(x_k) - f^*}{8mL} + \frac{\tau^2(m+8)}{64m} + \frac{\delta^2}{8\tau^2 L^2}. \tag{B.6}$$

Taking expectation with respect to $\mathcal{U}_{k-1}$ and defining $\rho_{k+1} \overset{\text{def}}{=} \mathbb{E}_{\mathcal{U}_k} r_{k+1}^2$ we obtain

$$\rho_{k+1} \leq \rho_k - \frac{\psi_k - f^*}{8mL} + \frac{\tau^2(m+8)}{64m} + \frac{D\delta}{4\tau L} + \frac{\delta^2}{8\tau^2 L^2}.$$

Summing up these inequalities from $k = 0$ to $k = M$ and dividing by $M + 1$ we obtain (5.2)

Estimate 5.2 also holds for $\hat{\psi}_M \overset{\text{def}}{=} \mathbb{E}_{\mathcal{U}_{M-1}} f(\hat{x}_M)$, where $\hat{x}_M = \arg\min_x \{f(x) : x \in \{x_0, \ldots, x_M\}\}$.

Now assume that the function $f(x)$ is strongly convex. From (B.6) we get

$$\mathbb{E}_{\xi_k} r_{k+1}^2 \overset{(B.4)}{\leq} \left(1 - \frac{\mu}{16mL}\right) r_k^2 + \frac{D\delta}{4\tau L} + \frac{\tau^2(m+8)}{64m} + \frac{\delta^2}{8\tau^2 L^2}$$

Taking expectation with respect to $\mathcal{U}_{k-1}$ we obtain

$$\rho_{k+1} \leq \left(1 - \frac{\mu}{16mL}\right) \rho_k + \frac{R\delta}{\tau L} + \frac{\tau^2(m+8)}{64m} + \frac{\delta^2}{8\tau^2 L^2}$$

and

$$\rho_{k+1} - \delta_\tau \leq \left(1 - \frac{\mu}{16mL}\right)(\rho_k - \delta_\tau) \leq$$

$$\leq \left(1 - \frac{\mu}{16mL}\right)^{k+1}(\rho_0 - \delta_\tau).$$

Using the fact that $\rho_0 \leq D^2$ and $\psi_k - f^* \leq \frac{1}{2} L\rho_k$ we obtain (B.5).

# C Missed proofs for Section 6

Let us define for any $\bar{x} \in \mathcal{E}$, $g \in \mathcal{E}^*$, $\gamma > 0$

$$x_X(\bar{x}, g, \gamma) = \arg\min_{x \in X} \left\{ \langle g, x \rangle + \frac{1}{\gamma} V(x, \bar{x}) + h(x) \right\}, \tag{C.1}$$

$$g_X(\bar{x}, g, \gamma) = \frac{1}{\gamma}(\bar{x} - x_X(\bar{x}, g, \gamma)). \tag{C.2}$$

We assume that the set $X$ is *simple* in a sense that the vector $x_X(\bar{x}, g, \gamma)$ can be calculated explicitly or very efficiently for any $\bar{x} \in X$, $g \in \mathcal{E}^*$, $\gamma$.

We will need the following two results obtained in [7].

**Lemma C.12.** *Let $x_X(\bar{x}, g, \gamma)$ be defined in* (C.1) *and $g_X(\bar{x}, g, \gamma)$ be defined in* (C.2). *Then, for any $\bar{x} \in X$, $g \in E^*$ and $\gamma > 0$, it holds*

$$\langle g, g_X(\bar{x}, g, \gamma) \rangle \geq \|g_X(\bar{x}, g, \gamma)\|^2 + \frac{1}{\gamma}(h(x_X(\bar{x}, g, \gamma)) - h(\bar{x})). \tag{C.3}$$

**Lemma C.13.** *Let $g_X(\bar{x}, g, \gamma)$ be defined in* (C.2). *Then, for any $g_1, g_2 \in E^*$, it holds*

$$\|g_X(\bar{x}, g_1, \gamma) - g_X(\bar{x}, g_2, \gamma)\| \leq \|g_1 - g_2\|_* \tag{C.4}$$

**Proof of Theorem 3.** First of all let us show that the procedure of search of point $w_k$ satisfying (6.3), (6.4) is finite. It follows from the fact that for $M_k \geq L$ the following inequality follows from (6.2):

$$\tilde{f}(w_k, \delta) - \frac{\varepsilon}{16M_k} \overset{(6.2)}{\leq} f(w_k) \overset{(6.2)}{\leq} \tilde{f}(x_k, \delta) + \langle \tilde{g}(x_k, \delta), w_k - x_k \rangle + \frac{L}{2}\|w_k - x_k\|_2 + \frac{\varepsilon}{16M_k}$$

which is (6.4).

Let us now obtain the rate of convergence. Using definition of $x_{k+1}$ and (6.4) we obtain for any $k = 0, \ldots, M$

$$f(x_{k+1}) - \frac{\varepsilon}{16M_k} = f(w_k) - \frac{\varepsilon}{16M_k} \overset{(6.2)}{\leq} \tilde{f}(w_k, \delta) \overset{(6.4)}{\leq} \tilde{f}(x_k, \delta) +$$

$$+ \langle \tilde{g}(x_k, \delta), x_{k+1} - x_k \rangle + \frac{M_k}{2}\|x_{k+1} - x_k\|^2 + \frac{\varepsilon}{8M_k} \overset{(C.2),(6.3)}{=}$$

$$= \tilde{f}(x_k, \delta) - \frac{1}{M_k}\left\langle \tilde{g}(x_k, \delta), g_X\left(x_k, \tilde{g}(x_k, \delta), \frac{1}{M_k}\right)\right\rangle + \frac{1}{2M_k}\left\|g_X\left(x_k, \tilde{g}(x_k, \delta), \frac{1}{M_k}\right)\right\|^2 + \frac{\varepsilon}{8M_k} \overset{(6.2),(C.3)}{\leq}$$

$$\leq f(x_k) + \frac{\varepsilon}{16M_k} - \left[\frac{1}{M_k}\left\|g_X\left(x_k, \tilde{g}(x_k, \delta), \frac{1}{M_k}\right)\right\|^2 + h(x_{k+1}) - h(x_k)\right] +$$

$$+ \frac{1}{2M_k}\left\|g_X\left(x_k, \tilde{g}(x_k, \delta), \frac{1}{M_k}\right)\right\|^2 + \frac{\varepsilon}{8M_k}.$$

This leads to

$$\psi(x_{k+1}) \leq \psi(x_k) - \frac{1}{2M_k}\left\|g_X\left(x_k, \tilde{g}(x_k, \delta), \frac{1}{M_k}\right)\right\|^2 + \frac{\varepsilon}{4M_k}.$$

for all $k = 0, \ldots, M$.

Summing up these inequalities for $k = 0, \ldots, N$ we get

$$\left\|g_X\left(x_{\hat{k}}, \tilde{g}_{\hat{k}}, \frac{1}{M_{\hat{k}}}\right)\right\|^2 \sum_{k=0}^{N}\frac{1}{2M_k} \leq \sum_{k=0}^{N}\frac{1}{2M_k}\left\|g_X\left(x_k, \tilde{g}_k, \frac{1}{M_k}\right)\right\|^2 \leq$$

$$\leq \psi(x_0) - \psi(x_{N+1}) + \frac{\varepsilon}{4}\sum_{k=0}^{N}\frac{1}{M_k}$$

Hence using the fact that $M_k \leq 2L$ for all $k \geq 0$ (which easily follows from the first argument of the proof) and that for all $x \in X$ $\psi(x) \geq \psi^* > -\infty$, we obtain

$$\left\|g_X\left(x_{\hat{k}}, \tilde{g}_{\hat{k}}, \frac{1}{M_{\hat{k}}}\right)\right\|^2 \leq \frac{1}{\sum_{k=0}^{N}\frac{1}{2M_k}}\left(\psi(x_0) - \psi^* + \frac{\varepsilon}{4}\sum_{k=0}^{N}\frac{1}{M_k}\right) \overset{M_k \leq 2L}{\leq}$$

$$\frac{4L(\psi(x_0) - \psi^*)}{N + 1} + \frac{\varepsilon}{2},$$

which is (6.5).

The estimate for the number of checks of Inequality 6.4 is proved in the same way as in [26].

## D   Table of notations

| | |
|---|---|
| $Q$ | Set of queries. |
| $V_q$ | Set of browsing graph vertices for query $q \in Q$. |
| $p_q$ | $\|V_q\|$ |
| $p$ | $\max_{q \in Q} p_q$ |
| $s_q$ | $\max_{i \in V_q} \|\{j : i \to j \in E_q\}\|$ – maximum number of neighbours in the graph |
| $s$ | $\max_{q \in Q} s_q$ |
| $V_q^j$ | Set of nodes annotated with label $\ell + 1 - j$. |
| $E_q$ | Set of browsing graph edges for query $q \in Q$. |
| $U_q \subset V_q$ | Seed set for query $q \in Q$. |
| $n_q$ | $\|U_q\|$ |
| $n$ | $\max_{q \in Q} n_q$ |
| $\mathbf{V}_i^q \in \mathbb{R}_+^{m_1}$ | Vector of node's features for $q \in Q$, $i \in V_q$. |
| $\mathbf{E}_{ij}^q \in \mathbb{R}_+^{m_2}$ | Vector of edges's features for $q \in Q$, $i \to j \in E_q$. |
| $\varphi := (\varphi_1, \varphi_2)^T$ | Vector of all parameters of the ranking algorithm. |
| $m$ | Total number of parameters, $m = m_1 + m_2$. |
| $\pi_q^0(\varphi)$ | Vector of restart probabilities for query $q \in Q$. |
| $P_q(\varphi)$ | Matrix of transition probabilities for query $q \in Q$. |
| $p_i(\varphi)$ | $i$-th column of the matrix $P_q^T(\varphi)$. |
| $\pi_q(\varphi)$ | Vector of page ranks given by solution of (2.3). |
| $\tilde{\pi}_q(\varphi, N)$ | Approximation for the vector $\pi_q(\varphi)$, obtained by Method (4.1). |
| $\pi_k$ | Intermediate iteration in Method (4.1). |
| $N$ | Number of steps in Method (4.1). |
| $\frac{d\pi_q(\varphi)}{d\varphi^T} \in \mathbb{R}^{p_q \times m}$ | Matrix of derivative of $\pi_q(\varphi)$ w.r.t. $\varphi$. |
| $\tilde{\Pi}_q(\varphi, N_2)$ | Approximation for the matrix $\frac{d\pi_q(\varphi)}{d\varphi^T}$, obtained by Method (4.4), (4.5). |
| $\Pi_0$ | $\alpha \frac{d\pi_q^0(\varphi)}{d\varphi^T} + (1-\alpha) \sum_{i=1}^{p_q} \frac{dp_i(\varphi)}{d\varphi^T} [\tilde{\pi}_q(\varphi, N_1)]_i$. |
| $\Pi_k$ | Intermediate iteration in Method (4.4), (4.5). |
| $N_1$ | Number of steps in Method (4.1) for approximation of matrix $\frac{d\pi_q(\varphi)}{d\varphi^T}$. |
| $N_2$ | Number of steps in Method (4.4), (4.5). |
| $A_q \in \mathbb{R}^{r_q \times p_q}$ | Matrix, representing assessor's view of the relevance of pages to the query $q \in Q$. |
| $r_q$ | $\sum_{1 \le j < l \le \ell} \|V_q^j\|\|V_q^l\|$. |
| $r$ | $\max_{q \in Q} r_q$. |
| $\alpha \in (0, 1)$ | Restart probability in the random walk on the browsing graph. |
| $R$ | Radius of the feasible set for the variable $\phi$. |
| $\hat{\varphi}$ | Center of the feasible set for the variable $\phi$. |
| $\Phi$ | $\{\varphi \in \mathbb{R}^m : \|\varphi - \hat{\varphi}\|_2 \le R\}$ – feasible set for the variable $\phi$ in the learning problem. |
| $f(\varphi)$ | The objective, which is minimized in the learning problem. |
| $\tilde{f}(\cdot, \delta_1)$ | Algorithmic approximation of $f(\cdot)$ with error $\delta_1$. |
| $\delta_1$ | Error of the algorithmic approximation for $f(\cdot)$. |
| $\tilde{g}(\cdot, \delta_2)$ | Algorithmic approximation of $\nabla f(\cdot)$ with error $\delta_2$. |
| $\delta_2$ | Error of the algorithmic approximation for $\nabla f(\cdot)$. |
| $g_\tau(\cdot, \delta)$ | Biased gradient-free oracle in Random Gradient-Free Method. |
| $\mathcal{E}, \mathcal{E}^*$ | Finite-dimensional real vector space, its dual. |
| $\langle g, x \rangle$ | Value of linear function $g \in \mathcal{E}^*$ at $x \in \mathcal{E}$. |
| $\|\cdot\|, \|\cdot\|_*$ | Norm on $\mathcal{E}$, its dual. |
| $L$ | Lipschitz constant of gradient of $f(\cdot)$ or inexact oracle parameter in (6.2). |
| $\delta$ | Parameter of inexact oracle in (6.2). |
| $\tau$ | Smoothing parameter in Random Gradient-Free Method. |
| $\xi$ | Random vector used in Random Gradient-Free Method. |
| $h$ | Stepsize in Random Gradient-Free Method. |
| $\varepsilon$ | Error of the solution of the learning problem. |
| $M$ | Number of steps on the upper level of Random Gradient-Free Method. |
| $d(x)$ | Prox-function used in gradient method. |
| $V(x, z)$ | Bregman distance used in gradient method. |
| $M_k$ | Appoximation for "Lipschitz constant" in gradient method at iteration $k$. |
| $L_0$ | Initial guess for "Lipschitz constant" in gradient method at iteration $k$. |