[Reviews · NeurIPS 2016]

Reviewer 1

Summary

This paper analyzes the convergence of a non-convex loss-minimization problem for learning the parameters of a general graph-based ranking model, that is defined by a random walk conducted by weights of nodes and edges, which are in turn defined by random walks defined by nodes’ and edge’s features. The optimization problem can not be solved by existing optimization methods which require exact values of the objective function. The proposed approach hence operates in two level. At the first level, a linearly convergent method is used to estimate an approximation to the stationary distribution of Markov random walk. This approach is validated among others and the authors show the value of the loss function can be approximated with any given precision. They also develop a gradient method for general constrained non-convex optimization problems using an inexact oracle, and prove its convergence to the stationary point of the problem. The contribution is in the adaptation of the approach to the case of constrained optimization problems when the value of the function can be calculated with some known precision. They prove the convergence of this method and exploit it on the second level of the proposed algorithm. The paper is well written and proofs seem to be correct though I did not go through all of them. My concern is that the main optimization points made here are adaptations of those proposed in “Yurii. Nesterov and Vladimir Spokoiny, Random Gradient-Free Minimization of Convex Functions, Foundations of Computational Mathematics, 2015, pp. 1–40”, and the supervised pagerank algorithm has been also proposed previously. On a minor level, a conclusion is expected especially that there is place to write one.

Qualitative Assessment

I would suggest to present the optimization problem in a more general context, showing the studied algorithm as to be a specific case. This would allow to better place the contribution with respect to the state-of-the-art.

Confidence in this Review

2-Confident (read it all; understood it all reasonably well)


Reviewer 2

Summary

This paper addresses formulate the Supervised PageRank models as a constrained non-convex optimization problem by incorporating properties of nodes and edges. The authors propose two approaches to solve the optimization problem. The first one is a gradient-free method with inexact zero-order oracle. The second one is a gradient-based method with inexact first-order oracle. My major complaint is that none of the experiment results are in the main text, which makes the paper incomplete.

Qualitative Assessment

Detailed comments. 1. The authors put experiment results in the supplement, which makes the main text incomplete. 2. Convexity is assumed in Theorem 1, while the function in (2.4) is non-convex. How to choose the set $\Phi$ to guarantee the convexity of the objective function? 3. Theorem 3 provides a bound for $M_K(x_K - x_{K+1})$. Could this bound ensure that the sequence $x_k$ converges to a stationary point? 4. By Figure 1 in the supplement, GFNM converges slowly. The authors may also want to report the efficiency of the algorithms.

Confidence in this Review

2-Confident (read it all; understood it all reasonably well)


Reviewer 3

Summary

In this paper, the authors proposed two optimization methods to solve the supervised PageRank problem. Unlike other existing methods, this paper proposed a gradient-based method with theoretically convergence rate guarantee and can achieve a given accuracy. The proposed gradient-free method has a guaranteed loss function decrease value. These two methods do not require exact value of the objective function and have a estimate of the convergence rate, also the performance is better than state-of-the-art in terms of the ranking quality. The hyper-parameters used in both methods are provided. The data sets are not publicly available but the description is clear so it may not be hard to reproduce their results.

Qualitative Assessment

The authors proposed two two-level optimization methods, namely gradient-based (GBN) and gradient-free (GFN) to solve the supervised PageRank problem. The lower level and upper level optimization is based on Nesterov and Nemirovski's work (ref:17). The proposed method does not require an exact value of the objective function and has proven estimate of the convergence rate given an accuracy. Therefore the proposed method can avoid computing the large matrix while still have a good result. From Section 2 to Section 5, the authors explained the details of their methods and included all the proofs in the supplementary. However, it may help a lot if the authors can provide a symbol table since there are too many symbols in their equations and it is really easy to lose track of them. For example in Line 139 and equation 3.1 the authors used $N$ and they did not mention the meaning of it (number of steps) until Line 193. Also, in the experiment section, the authors only compared their work with GBP regarding to the loss function instead of the ranking quality. Moreover, since this work is a generalization of ref:17, I think it would be good if the authors can compare with it directly so we can see if the performance is better.

Confidence in this Review

1-Less confident (might not have understood significant parts)


Reviewer 4

Summary

This paper propose gradient-based and random-gradient-free methods to solve a non-convex loss-minimization problem of learning supervised PageRank models.

Qualitative Assessment

It is better to report the time of the proposed optimization method.

Confidence in this Review

1-Less confident (might not have understood significant parts)


Reviewer 5

Summary

The authors motivate pagerank and then bring up a framework which has node and edge weights from a feature model. They pointed out that a previous algorithm for learning the parameters of the model had no performance guarantees. They plan to solve the methods by combining a previous method for approximating the stationary distribution of a Markov random walk with an adaptation of a gradient free method used for optimization when the loss function can only be estimated. They reproduce a framework under a different name and notation from [21]. They discuss a number of technical assumptions and a loss function and explain why the problem is hard to solve. Some lemma bounding the approximate gradient are stated. A general gradient free method is discussed, using a biased gradient-free oracle, and state a theorem on its performance. They make some more assumptions and construct an algorithm for the learning problem and adapt the gradient free methods to their problem, stating some more bounds. They outline some details of experimental results on a web graph used in related work, but detailed reporting of results and analysis are relegated to the appendix. Some parameters are not reported, and the novel algorithm is proclaimed to have outperformed other algorithms, but there is no discussion of the extent or implication of the improvement. It is unclear whether a diverse set of graphs were considered for testing.

Qualitative Assessment

The writing is littered with grammatical errors. Citations are sloppy, the problem is called Supervised PageRank with a cite to [21], but only semi-supervised page rank appears there. I am not convinced that the problem solved is impactful. Your bounds are strong, but I have doubts about whether the technical assumptions stated will hold in emprical data. Some assumptions seem limiting, e.g. your constriction of the search space of phi to an R-ball around some chosen phi_0may greatly restrict the efficacy/applicability of your algorithm, which should be discussed. My biggest issue with the paper is that it all the meat appears in the appendix. You neither provide proof outlines nor full empirical results. How did you choose phi_0 and R in the empirical results section? Did you run tests on more than one graph? You do not report any tables or specific results in the main body of the paper. What are the practical consequences of your improvements? Your bounds and analysis and algorithm may be powerful, but the content in the paper itself fails to convince me.

Confidence in this Review

1-Less confident (might not have understood significant parts)